# Modelled Effects of Temperature Gradients and Waves on the Hydroxyl Rotational Distribution in Ground-Based Airglow Measurements

Christoph Franzen[1,2], Patrick Joseph Espy[1,2], Robert Edward Hibbins[1,2]

[1] Norwegian University of Science and Technology (NTNU), Trondheim, 7491, Norway
[2] Birkeland Centre for Space Science (BCSS), Norway

*Correspondence to*: Christoph Franzen (franzen.christoph@rwth-aachen.de)

**Abstract.** Spectroscopy of the hydroxyl (OH) airglow has been a commonly used way to remotely sense temperatures in the mesopause region for many decades. This technique relies on the OH rotational state populations to be thermalised through collisions with the surrounding gas into a Boltzmann distribution characterised by the local temperature. However, deviations of the rotational populations from a Boltzmann
distribution characterised by a single temperature have been observed and attributed to an incomplete thermalisation of the OH from its initial, non-thermodynamic equilibrium distribution. Here we address an additional cause for the apparent amount of excess population in the higher rotational levels of the OH airglow brought about by integrating these OH emissions through vertical gradients in the atmospheric temperature. We
find that up to 40 % of the apparent excess population, currently attributed to incomplete thermalisation, can be due to the vertical temperature gradients created by waves. Additionally, we find that the populations of the different upper vibrational levels are affected differently. These effects need to be taken into account in order to assess the true extent of non-thermodynamic equilibrium effects on the OH rotational populations.

## 1 Introduction

The hydroxyl (OH) airglow has been employed for many years for remote sensing of the Mesosphere and Lower Thermosphere (MLT) region, an example of which may be found in Smith et al. (2010). The 8-km thick airglow layer is created at about 90 km altitude (Baker and Stair, 1988; Xu et al., 2012) by the highly exothermic reduction of ozone:

$$H + O_3 \rightarrow OH^* + O_2 \ (5.3 \cdot 10^{-19} J) \tag{1}$$

The excess heat of reaction, ~$5.3 \cdot 10^{-19}$ J, produces the $OH^*$ in excited vibrational quantum levels of $v'$=6-9 (e.g.
Mlynczak and Solomon (1993)). Subsequent radiative cascading and collisional-deactivation produces $OH^*$ in all vibrational levels $\leq 9$. Radiative de-activation can occur between any two vibrational quantum levels, but transitions with $\Delta v = 2$ are preferred (Langhoff et al., 1986).

The excess energy of reaction (1) also creates rotational excitation within the $OH^*$ molecule in addition to the vibrational excitation. The nascent rotational population for high rotational levels, $J'$, shows a distribution
characteristic of a temperature far above the local atmospheric temperature. Llewellyn & Long (1978) reported a nascent temperature of 760 K for $v'=$ 9, whereas others report temperatures as high as 9000-10000 K (Dodd et al, 1993; Oliva et al., 2015; Kalogerakis et al., 2018). Low rotational levels (with $N \leq 4$) with energy separations less than $kT$, the amount typically exchanged during collisions, have been observed to have efficient energy transfer in the thermalisation process (Maylotte et al., 1972; Polanyi and Sloan, 1975; Polanyi and Woodall, 1972). Thus,

emission from these states has been observed to be characterised by a single temperature Boltzmann distribution (Harrison et al., 1971; Harrison et al., 1970; Pendleton et al., 1993; Perminov et al., 2007; Sivjee et al., 1972; Sivjee and Hamwey, 1987). However, emission observed from the higher rotational levels (N>4), where the energy separation exceeds $kT$, has indicated an anomalous, non-thermalized population that cannot be described using the same Boltzmann temperature that characterizes the lower rotational levels (Cosby and Slanger, 2007; Dodd et al., 1994; Kalogerakis, 2019; Noll et al., 2015; Pendleton et al., 1989; Pendleton et al., 1993). In keeping with the terminology employed by Pendleton et al. (1993), we refer to the condition where the population can be described by a single-temperature Boltzmann distribution as Local Thermodynamic Equilibrium (LTE), whereas Non-Local Thermodynamic Equilibrium (NLTE) is used when the population departs from that distribution.. Work is currently underway to use observations of the excess populations in the high rotational levels of the OH airglow to determine state-to-state quenching coefficients and to understand the thermalisation process in OH (Kalogerakis et al., 2018). The term LTE as used here is not technically correct as it does not account for radiative effects (i.e. the emission of airglow photons) on the rotational-vibrational level population distribution. Instead it relates only to the collisional distribution of the rotational levels being characterized by the temperature of the surrounding gas, as has been done in Pendleton et al. (1993).

Here we examine the effects of temperature gradients in the OH emission region on the resulting vertically integrated spectrum of the Meinel Δv=2 sequence. To achieve this, model work was executed, where the model assumes that for each vibrational level, the rotational population distribution of the OH is in LTE at every altitude. The emission in each rotational line is then integrated vertically. We find that even if the OH rotational levels are in strict LTE with the surrounding atmosphere, the temperature gradients through the OH emission region will create apparent excess emission in the higher OH rotational lines. Here we calculate the apparent excess population relative to the Boltzmann population expected using the temperature determined by the population of rotational levels with $N \leq 4$. This excess population can be incorrectly interpreted as due to NLTE effects, affecting the subsequent calculations of the thermalisation process. The deviations in the inferred populations from a single-temperature Boltzmann distribution are compared with observations that include both NLTE and temperature gradient effects. This comparison is made for realistic atmospheric temperature profiles that have been perturbed with realistic atmospheric gravity-waves in order to help quantify the true NLTE content needed to construct a quantitative picture of OH thermalisation. Recent measurements indicate that the Boltzmann distribution of the rotational levels may be characterized by a temperature that is higher than that of the surrounding gas due to incomplete thermalization (Noll et al., 2018). However, the purpose of this paper is to show that even if complete thermalization with the surrounding gas takes place, ground based measurements integrating through temperature gradients within the OH layer will not see a rotational population described by a Boltzmann distribution characterized by a single temperature.

## 1.1 Overview of the OH emission

Fig. 1 shows an example of the OH airglow volume emission rate (VER) and a temperature profile, measured from the Sounding of the Atmosphere using Broadband Emission Radiometry (SABER) instrument aboard the NASA Thermosphere Ionosphere Mesosphere Energetics Dynamics (TIMED) satellite (Mlynczak, 1997; Russell et al., 1999). This specific measurement is a zonal mean and monthly average from July 2016 between the latitudes of 20 and 30 degrees north. The OH VER from the vibrational levels 8 and 9 is shown in red in the figure, while

the black curve shows the temperature profile. It can be seen that the temperature is not constant through the OH layer. In this example the atmospheric temperature changes by over 10 K through the layer. This observed behaviour is similar to other observations, for example from French and Mulligan (2010), who compared TIMED/SABER observations with ground-based observations.

5   Waves will exacerbate this effect by perturbing both the OH VER and changing the temperature gradient. Thus, the rotational level population distribution of the OH, even if thermalised at each altitude, will have different temperatures at each of those altitudes. Any instrument that integrates through the OH layer will therefore not see rotational line emission resulting from a single, average temperature, but from the whole span of temperatures present in the layer..

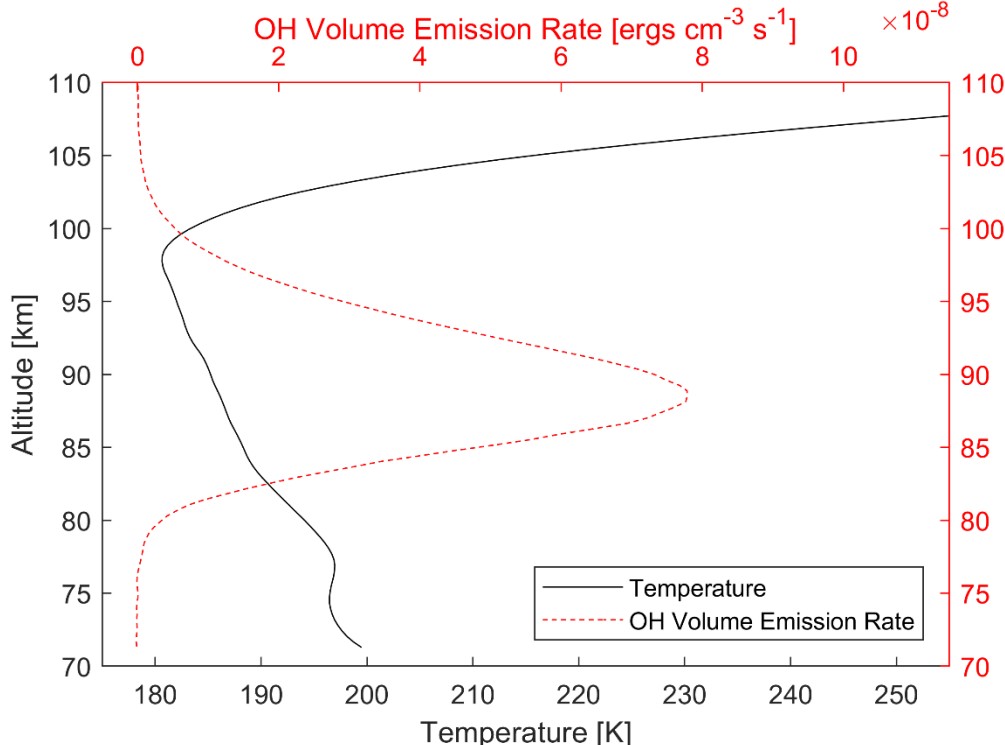

Figure 1: Example of a typical temperature profile (black) of the airglow layer and the VER (red-dashed) from the SABER satellite showing the variation of temperature through the emission region. This specific measurement is a zonal mean and monthly average from July 2016 between the latitudes of 20 and 30 degrees north for the $\Delta v=2$ bands from the 8 and 9 upper-state vibrational levels.

15   **2 Method**

We utilise a steady-state model of the OH VER, described below, to synthesise individual synthetic rotational spectra at 1 km intervals from 74 to 110 km. The model assumes that for each vibrational level, the rotational level population distribution of the OH is in LTE (i.e. can be described by a single-temperature Boltzmann distribution) with the local temperature at each altitude. Each rotational line is integrated in altitude to give the

20   net spectrum that would be observed by an instrument integrating through the layer. The distribution of emission in the rotational lines is then used to infer the population of the OH rotational levels, allowing us to quantify the

portion of the inferred excess population in the upper levels that is due to the temperature gradients across the OH layer.

## 2.1 The OH steady state model

The atmospheric background temperature and concentrations of $N_2$, $O_2$, H and O are taken from the US Naval Research Laboratory's Mass Spectrometer and Incoherent Scatter radar model (NRLMSISE-00) (Picone et al., 2002). The steady-state ozone concentration is then calculated from balancing the production and loss processes. The production mechanism is:

$$O + O_2 + M \rightarrow O_3 + M, \tag{2}$$

where M is a reaction mediator. The temperature dependent rate coefficients for this reactions are taken from the International Union for Pure and Applied Chemistry (IUPAC) Gas Kinetic Database (Atkinson et al., 2004). Loss processes include losses due to O via

$$O + O_3 \rightarrow 2O_2, \tag{3}$$

using the reaction rate coefficient of Sander et al. (2003). The loss of $O_3$ to atomic hydrogen,

$$H + O_3 \rightarrow OH^* + O, \tag{4}$$

was also used to calculate the production rate of $OH^*$ for each vibrational level, using the reaction rate coefficient (Sander et al., 2003). Due to the exothermicity of reaction 4, vibrational levels from v'=6 to 9 can be populated. The production of each vibrational level $OH^*(v')$ is calculated using the branching ratios from Sander et al. (2019). Collisional loss for each $OH^*(v')$ vibrational level was calculated for collisions with O, $O_2$ and $CO_2$ using the rate coefficients of Dodd et al. (1991), Knutsen et al. (1996), Dyer et al. (1997) and Chalamala and Copeland (1993). The model assumes quenching to the ground vibrational state, known as "sudden death" for the O, and stepwise quenching by one vibrational unit for the $O_2$ and the $CO_2$ (McDade and Llewellyn, 1987). $N_2$ is not considered as a quencher in this model. The rate coefficient for OH quenching with $N_2$ is small, and (Knutsen et al., 1996) were only able to provide an upper limit. Since the $O_2/N_2$ mixing ratio is nearly constant up to the turbopause, the $O_2$ is three times more effective at quenching than the $N_2$. Thus, neglecting the $N_2$ is well within the uncertainty of the $O_2$ rate coefficient and does not significantly affect the altitude distribution of the OH. The relative shape and peak height of the altitude profiles of the individual vibrational levels agree closely with those of the more sophisticated model of Adler-Golden (1997)" As previously mentioned, the rotational population distribution within this vibrational level is taken to be a Boltzmann distribution characterized by the local temperature at this altitude.

The total radiative loss from each vibrational level is given by $N_{v'}(z) \cdot A_{v'}$, where $N_{v'}(z)$ is the concentration of the hydroxyl $v'$ vibrational level at altitude z and $A_{v'}$ is its inverse lifetime, calculated from (Langhoff et al., 1986). The total VER of any $v'$ to $v''$ vibrational transition is then given by $V_{v'v''}(z) = A_{v'v''} \cdot N_{v'}(z)$, where $A_{v'v''}$ is the transition probability for the vibrational transition from $v'$ to $v''$ that is calculated from the Einstein coefficients from (Langhoff et al., 1986). Although transition probabilities from a number of different studies differ from those of Langhoff et al. (1986), the integration over the rotational distribution, required to obtain the vibrational

population, is relatively insensitive to the choice of transition probabilities. This radiative cascade into lower vibrational levels then acts as an additional production term for levels below v'=9. Balancing these production and loss terms at each height yields the steady-state concentration of each OH* vibrational level as a function of height, $N_{v'}(z)$. Such a model has been shown to fit observations, for example TIMED/SABER (Xu et al., 2012). The application of this model to the simulation of a ground-based measurement under the influence of a realistic temperature gradient is undertaken as described below.

### 2.2 The wave model

As mentioned above, the background atmospheric temperature profile can be perturbed by waves. We therefore include wave-induced temperature perturbations in the model. Following Holton (1982), we limited the wave growth with altitude to maintain temperature gradients to below the dry adiabatic lapse rate. The wave has the form

$$A_w(z) \cdot \cos\left(\frac{2\pi z}{\lambda_w} + \varphi_w\right), \tag{5}$$

Where $\varphi_w$ is the wave's phase, and $A_w(z)$ is a function of altitude so as not to exceed the dry adiabatic lapse rate. The lower edge of the model is 74 km altitude. The wave amplitude as a function of altitude is shown in Fig. 2 for the case of a wave with an amplitude of 10 K at 74 km altitude and a vertical wavelength of 20 km. This example is given for an isothermal atmosphere of 200 K (Fig. 2 (a), dashed-black). The wave grows in amplitude with altitude to conserve energy (dashed-blue), but at regions where the lapse rate exceeds the dry adiabatic lapse rate (here between 90 and 95 km, and between 110 and 115 km), the wave loses energy and the amplitude decreases (Holton, 1982). The breaking wave is shown in red in Fig. 2 (a). Fig. 2 (b) shows the instantaneous lapse rate (change in temperature with altitude) of the non-breaking wave (dashed-blue) and the breaking wave (red), which never crosses the dry adiabatic lapse rate of 10 K/km (dashed-black). Fig. 2 (c) shows the amplitude of the non-breaking wave (dashed-blue) which increases exponentially and for the breaking wave (red), which decreases at the altitudes where the wave dissipates energy.

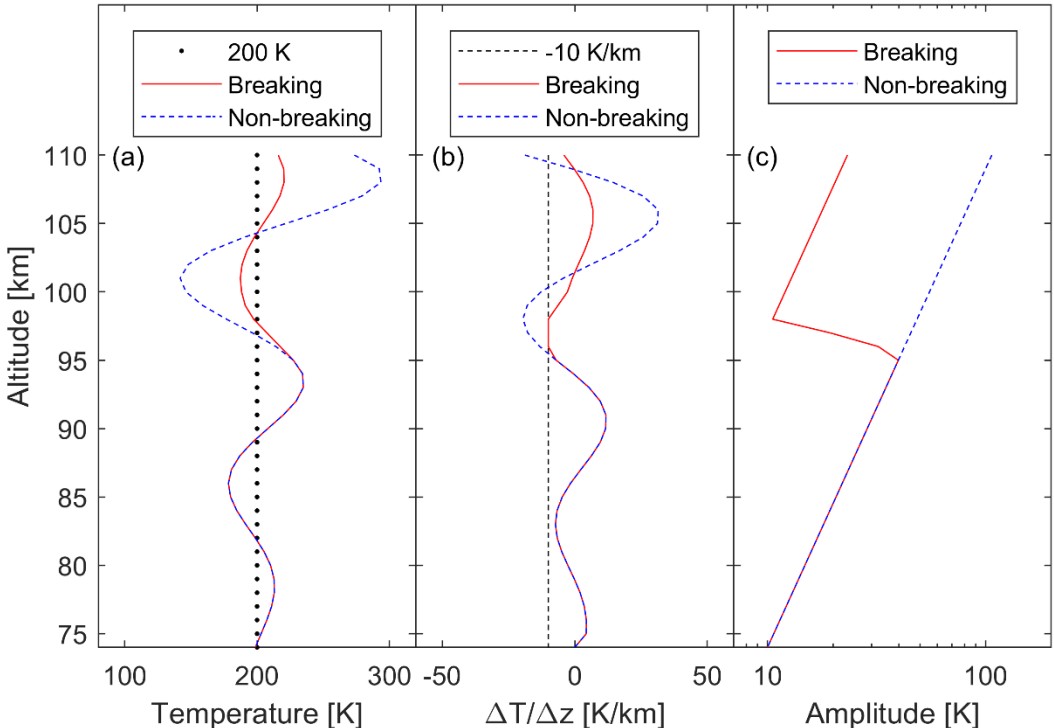

Figure 2: (a) Example of a wave of amplitude 10 K at 74 km altitude perturbing an isothermal atmosphere at T = 200 K (dotted-black). The dashed-blue line represents the wave without any breaking, and the red line is the breaking wave as used in the model. (b) The rate of change of temperature with altitude for the wave shown in (a). When the temperature changes faster than 10 K/km (dashed-black) the wave breaks. (c) At these altitudes, the amplitude of the breaking wave (red) decreases, while the non-breaking wave continues to grow exponentially (dashed-blue).

The background atmosphere mixing ratios from the NRLMSISE-00 model were also perturbed using the gravity wave polarisation relations from Vincent (1984). This perturbed background atmosphere and temperature profile were then used in the steady-state model for the OH to yield a new, wave-perturbed $N_{v'}(z)$ for the analysis.

### 2.3 Simulation of a ground-based measurement

The VER of a rotational transition for the $J'$ to $J''$ state from the upper state, $v'$, of the v' to v" vibrational band is given by:

$$V_{v'v''J'J''}(z) = N_{v',J'}(z) \cdot A_{v'v'',J'J''} , \tag{6}$$

where $A_{v'v'',J'J''}$ is the transition probability for the $J'$ to $J''$ rotational transition in the $v'$ to $v''$ vibrational transition. Due to vibrational-rotational coupling, these coefficients are specific to each vibrational transition. $N_{v',J'}(z)$ is the population of the upper rotational level, $J'$ in the upper vibrational level, $v'$. We concentrate in this paper on the 3/2 electronic subset of the OH airglow and drop the spin-orbit splitting quantum number F in all equations for readability. Assuming that collisions have thermalised the closely spaced rotational levels with the surrounding gas at temperature $T$, their population may be described using a Boltzmann distribution written as:

$$N_{v',J'}(z) = \frac{N_{v'}(z) \cdot 2(2J' + 1)}{Q_R(T(z))} \cdot exp\left(\frac{-E_{v',J'}}{k_B \cdot T(z)}\right) \tag{7}$$

Where $N_{v'}(z)$ is the total population of the $v'$ vibrational level at altitude z calculated from the model, $E_{v',J'}$ is the energy of the $J'$ rotational level, the factor $2(2J'+1)$ is the degeneracy of that level (including Λ-doubling) and $Q_R(T(z))$ is the rotational partition function (Herzberg, 1950). $k_B$ is the Boltzmann constant.

Using the rotational transition probabilities of Rothman et al. (2013), the rotational line VERs are calculated within a single $v'$ to $v''$ transition assuming that the OH is in LTE for the rotational level populations with the surrounding gas at each altitude and therefore follows a Boltzmann distribution of population characterised by the local temperature. The VER of each rotational line of a given vibrational transition is integrated through the layer from 74 km through to 110 km to give the intensity of the line, $I_{v'v'',J'J''}$. This results in a net spectrum of low rotational lines whose intensities are enhanced relative to the mean temperature of the emission region when emitted from the cooler regions. Accompanying this are high rotational lines whose intensities are enhanced relative to the mean temperature of the emission region when emitted from the warm regions. Each of these is weighted by the VER of the vibrational transition at each altitude, $N_{v'}(z) \cdot A_{v'v''}$.

As an example, the OH model was used to create a synthetic spectrum of the (7,4) rotational-vibrational band, assuming LTE for the rotational level populations at every altitude level, for conditions for mid-July for a mid-latitude region. Specifically, the data presented are for Boulder, Colorado (40.0° N; 105.6° W) to make a direct comparison with the findings from Pendleton et al. (1993). The background temperature profile (black line) and the (7,4) VER variation with altitude (red line) are shown in Fig. 3 (a). These are perturbed by the wave with an initial amplitude of 10 K at 74 km altitude, which has grown to about 30 K at 90 km altitude, and a vertical wavelength of 30 km, shown in blue-dashed. Waves with similar amplitudes have been observed at these altitudes (Picard et al., 2004).

The resulting distribution of rotational line intensities as a function wavelength for two altitudes is shown in Fig. 3 (b) and (c). The net spectrum as observed by a ground-based instrument is analysed, as detailed below, to examine the influence of this high-rotational level tail on the fitted temperature.

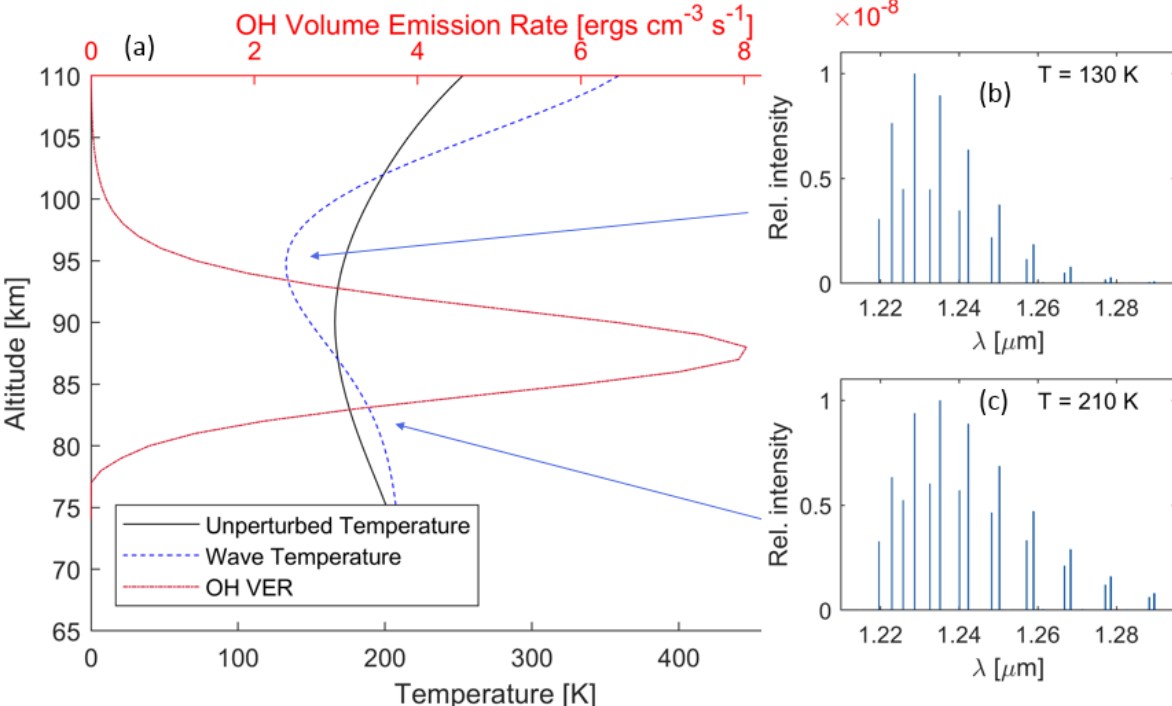

Figure 3: (a) Illustration of the change in the OH airglow spectrum with altitude due to the temperature profile. The red line shows a modelled VER profile of the OH in ergs $cm^{-3}$ $s^{-1}$. The black line shows the background temperature profile retrieved from the NRLMSISE-00 model. The blue dashed line shows the same temperature profile but perturbed by a wave with an amplitude which has grown to about 30 K at 90 km altitude and a vertical wavelength of 30 km. The insets (b) and (c) on the right show OH spectra of the (7,4) transition at different altitudes for the wave case. Note that even though the figure shows the $P_1$ and $P_2$ lines, only the $P_1$ lines are considered in the model. With the example wave given here, the temperature variation increases the VER of the lower rotational lines at 95 km altitude, and of the higher rotational lines at 80 km.

## 2.4 Temperature fitting

After integration through the layer, the relative population of the $J'$ state relative to the lowest rotational energy level, $E_{v'}$, is now given in terms of the line intensity by:

$$\frac{I_{v',v'',J'J''}}{2(2J'+1) \cdot A_{v'v'',J'J''}} = Q_R(T)N_{v'} \cdot exp\left(\frac{-\left(E_{v',J'} - E_{v'}\right)}{k_B T}\right) \tag{8}$$

Where $N_{v'}$ is the integral of the vibrational population $N_{v'}(z)$ over altitude, and $T$ is the effective rotational temperature of the altitude-integrated spectrum. From this, the observed line intensities from rotational levels of known quantum number, energy, and transition probability may be used to define the relative total population and temperature using:

$$\ln\left(\frac{I_{v',v'',J'J''}}{2(2J'+1) \cdot A_{v'v'',J'J''}}\right) = \ln(Q_R(T)N_{v'}) - \frac{1}{T}\left[\frac{\left(E_{v',J'} - E_{v'}\right)}{k_B}\right] \tag{9}$$

Fig. 4 shows the result in terms of Equation (9), of integrating each rotational line in the (7,4) Meinel band for the atmospheric perturbation of a wave with an amplitude of 30 K at 90 km altitude and 30 km vertical wavelength. This is the same wave as shown in Fig. 3. Following Pendleton et al. (1993), fitting a temperature to the lowest three rotational levels of the OH spectrum yields the dashed red curve shown in Fig. 4. It may be seen that the lowest three rotational levels are well characterized by a single, Boltzmann rotational temperature, $T_{1,3} = 155.8 \pm 0.8$ K, as has been observed in nightglow spectra (Espy and Hammond, 1995; Franzen et al., 2017; Harrison et al., 1970; Noll et al., 2015). However, there is excess emission in the higher rotational lines which could be interpreted as populations exceeding that expected from a thermalised Boltzmann distribution. This apparent excess population occurs even though the OH distribution was constrained to be a Boltzmann distribution with a single temperature at each altitude.

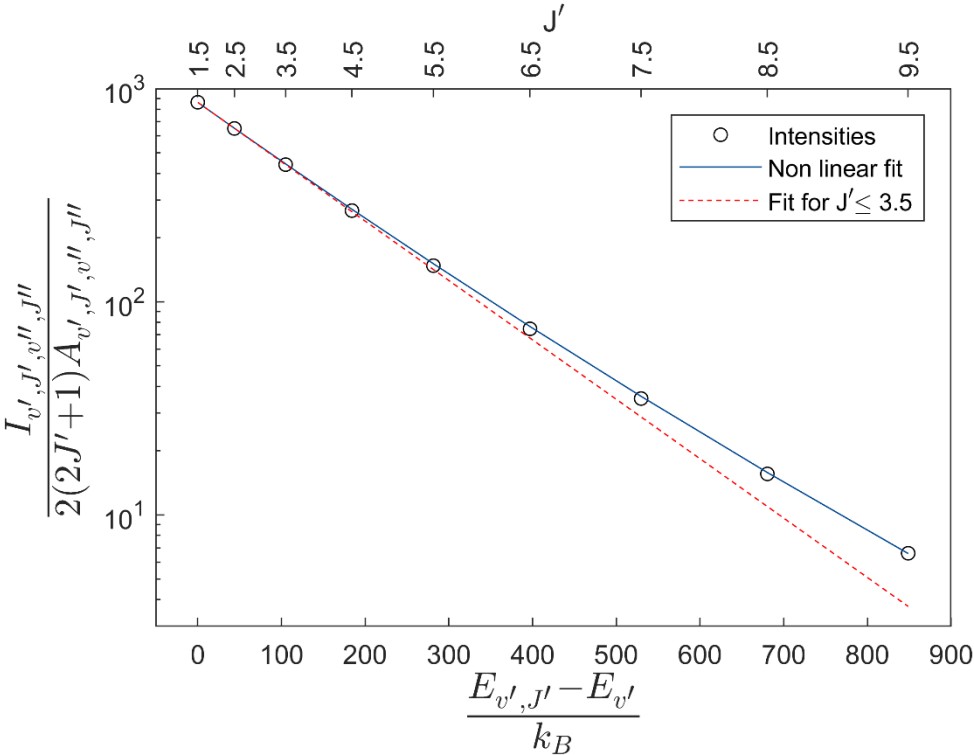

Figure 4: Population of P-branch lines of the (7,4) Meinel band as calculated with an atmospheric background profile containing the wave shown in Fig. 3. The energies given on the x-axis are relative to the lowest rotational energy. An exponential fit to the lowest three rotational levels (red dashed line) underestimates the populations at higher levels. A non-linear fit as presented in Equation (10) can fit the intensities of rotational levels as high as $J'$ = 9.5, while reproducing the same temperature as the fit to the lowest three levels.

A non-linear formulation can be used in order to characterise this excess population. This non-linear fit is of the form:

$$\ln\left(\frac{I_{v',v'',J'J''}}{2(2J'+1)\cdot A_{v'v'',J'J''}}\right) = \ln(Q_R(T)N_{v'}) - \frac{1}{T}\left[\frac{(E_{v',J'} - E_{v'})}{k_B}\right] + \frac{\beta}{T^2}\left[\frac{(E_{v',J'} - E_{v'})}{k_B}\right]^2 \qquad (10)$$

Here $\beta$ is a non-linearity parameter, which is a free parameter in the non-linear fit. This non-linear fit is also presented in Fig. 4 as a solid blue line, showing that the intensity in rotational lines as high as $J' = 9.5$ is now fitted. The retrieved temperature $T_{NL} = 154.9 \pm 0.1$ K, is the same as $T_{1,3}$ within the fitting uncertainties, with a $\beta$ factor of 0.019.

## 2.5 Expansion to arbitrary waves

So far, we have considered only a single wave with one given amplitude, wavelength and phase as an example. Different wave amplitudes $A_w$ will, of course, change the populations seen in Fig. 4. In the limiting case, where $A_w = 0$, the original background temperature profile is obtained. Different wavelengths $\lambda_w$ will also change the shape of the atmospheric temperature profile. Short $\lambda_w$ waves can change the temperature toward both higher and lower temperatures within the OH layer. Thus, their effect can be small when integrated over the whole layer. Longer $\lambda_w$ waves, especially with $\lambda_w$ on the order of, or longer than, the thickness of the OH layer, can introduce a temperature change that only warms one end of the layer and cools the opposite end. When such gradients reinforce the background temperature change with altitude, they can change the total integral over the OH layer substantially, as seen in Fig. 4. Lastly, the wave phase $\varphi_w$ can change the influence of the wave on the total integral of the OH layer. The same wave would have the opposite temperature perturbation if $\varphi_w$ was shifted by half a wavelength. In this case, the wave would work against the natural temperature gradient of the background atmosphere without a wave, resulting in a smaller effect than that shown in Fig. 4.

All three wave parameters - $A_w$, $\lambda_w$ and $\varphi_w$ - should therefore be considered when modelling different waves to examine their influence on the total integrated OH spectrum. In this research, the three parameters were adjusted in equidistant steps. The amplitude $A_w$ was varied between 0 K and 40 K at an altitude of 90 km, spanning a range of previously observed semi-diurnal tide amplitudes (Hagan et al., 1999; Oberheide et al., 2011; Picard et al., 2004; She et al., 2002; Shepherd and Fricke-Begemann, 2004; Zhang et al., 2006). The vertical wavelength $\lambda_w$ was varied between 2 km and 80 km, spanning the range of gravity waves, tides and planetary waves (Davis et al., 2013). The phase $\varphi_w$ was varied between 0 and $2\pi$.

## 3 Results and discussion

Fig. 4 shows that a large atmospheric temperature gradient can produce higher populations in the higher $J'$ rotational lines than would be expected from a strict LTE fit with only one effective temperature. We now want to quantify how large this apparent excess population can become and compare it to the study by Pendleton et al. (1993). An ensemble of waves was simulated as described above. The apparent excess population and the non-linearity parameter, $\beta$, were calculated for each wave-perturbed temperature background profile.

Fig. 5 shows two different wave scenarios for the (7,4) transition. The red plot (with the axis on the left-hand side) shows a no-wave scenario, where the atmospheric background temperature profile from the NLRMSISE-00 model is used. The apparent excess population is the ratio between the intensity of a rotational line integrated in altitude, and the intensity of that line predicted by a Boltzmann distribution fitted to the distribution of integrated line intensities of the lowest three rotational lines using a single, effective temperature. Thus, an apparent excess population of 1 is the same population as predicted from a single-temperature Boltzmann distribution fitted to the lowest three rotational lines. Similarly, an apparent excess population of 2 is a population twice as large as that predicted from this single-temperature Boltzmann distribution. It is clear that the first three lines with $J' \leq 3.5$ can

be characterised by a single, effective Boltzmann temperature. However, all higher lines show populations in excess of that expected from a single effective temperature. The effective overpopulation approaches approximately 1.12 for $J' = 9.5$, even though the OH molecule is thermalised with the local temperature at every altitude.

5 The blue bars (with the plot on the right side) show a scenario with a wave perturbing the atmosphere. This specific wave has an amplitude $A_w = 30$ K at 90 km altitude, $\lambda_w = 30$ km and $\varphi_w = 4.9$ rad. This would be a large amplitude gravity wave, but still within the range of atmospheric tidal observations. Although different $J'$ levels are affected differently, the general shape is similar to that observed with the no-wave scenario. However, the magnitude of the effect is much greater. The first three lines show the populations expected in a single temperature

10 LTE case, while higher $J'$ levels yield increasingly higher apparent excess populations. For the highest line considered in this paper with $J' = 9.5$, the apparent excess population is twice that expected for a Boltzmann distribution with a single effective temperature, despite the OH being in LTE for the rotational level populations at every altitude.

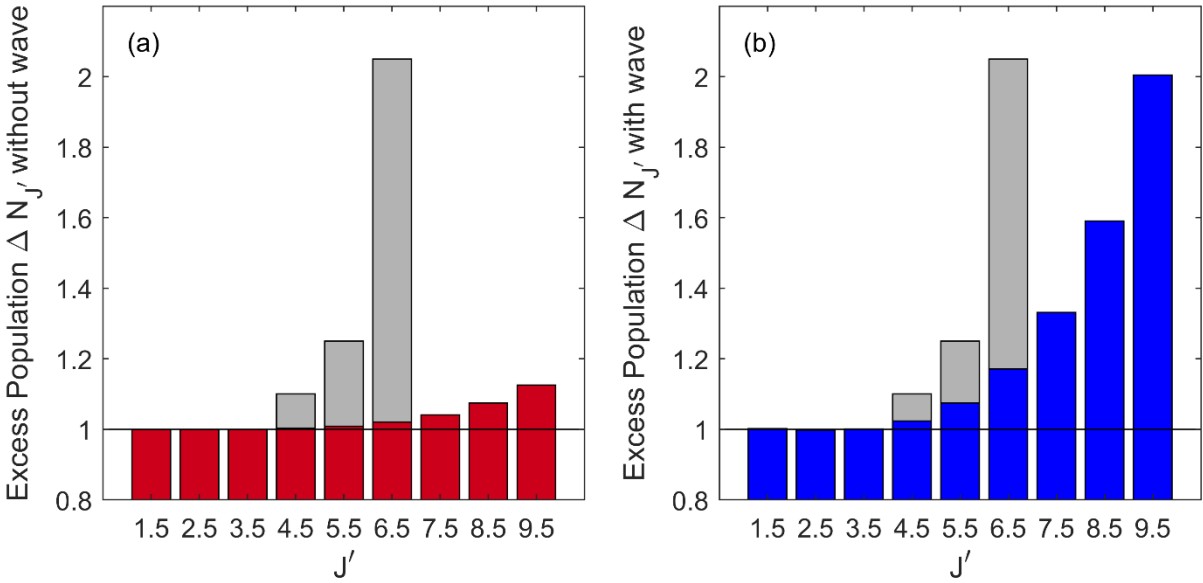

Figure 5: The calculated apparent excess population of the OH (7,4) P($J'$) lines relative to the Boltzmann population of the fitted temperature to the lowest three lines. Red on the left is calculated with the climatological temperature gradient shown from NRLMSISE-00 above Boulder, Colorado (40.0° N; 105.6° W) in mid-July. Blue on the right is calculated with a wave of $A_w = 30$ K at 90 km, $\lambda_w = 30$ km and a phase that yields the maximum

20 effect, $\varphi_w = 4.9$ rad. This is the same wave as presented in Fig. 3 and 4. The distribution has a $\beta = 0.019$. The grey bars represent the measurements ascribed to NLTE effects from Pendleton et al. (1993).

These results can be compared to the findings from Pendleton et al. (1993) in their Fig. 16, for the (7,4) transition above Boulder, Colorado during mid-summer, the same season and transition as presented here. These results are shown as gray bars in Fig. 5 (a) and (b). Although the measurement from Pendleton et al. (1993) included NLTE

25 effects, the overall shape of the distribution is similar to the LTE for the rotational level populations simulations

presented here. When comparing these results, note that Pendleton et al. (1993) use the lower-state quantum number $N$, which corresponds to our $J' + \frac{1}{2}$. The first three rotational levels in this study and in Pendleton et al. (1993) have populations that follow a Boltzmann distribution characterized by a single temperature, and above that, there is excess population. The difference between the absolute numbers we observe and those of Pendleton et al. (1993) indicates the portion of the NLTE that might be due to the temperature gradient effects observed here. For example, Pendleton et al. (1993) reported an apparent excess population of around a factor of 2 for the P($N$=7) (i.e. $J'$=6.5) line. However, for the two cases presented here in Fig. 5, there is less apparent excess population. While the no-wave scenario yields an apparent excess population of 1.02, a background profile with a wave yields 1.17 times the population. These numbers mean that up to about 17 % of the effect Pendleton et al. (1993) observed could be due to wave activity and not NLTE effects. Thus, the effect of the atmospheric temperature background has to be considered in addition to NLTE effects whenever the populations inferred from integrated airglow observations of high $J'$ lines are to be used in kinetic thermalisation studies.

This calculation of the apparent excess population can now be repeated for waves of different amplitudes, wavelengths and phases. Fig. 6 shows the non-linearity in terms of the $\beta$-value of the temperature fit (from Equation (10)) and the corresponding apparent excess population of the sixth rotational line with $J' = 6.5$ ($N = 7$), the highest line reported by Pendleton et al. (1993) in their NLTE study of the OH. The figure shows results related to the (7,4) vibrational transition, which is the same transition that Pendleton et al. (1993) used. Fig. 6 (a) shows that for vertical wavelengths above about 20 km, the non-linearity of the temperature fit increases with wave-strength, approaching 2% of the linear temperature variation (see Equation 10). The maximum non-linearity is observed at vertical wavelengths of about 20 km for waves with an amplitude of below 10 K, while for waves with an amplitude of 40 K the maximum non-linearity is observed at vertical wavelengths around 40 km. For shorter vertical wavelengths, the non-linearity is smaller (at about 0.5 %) and decreases with increasing wave amplitude. Fig. 6 (b) shows the apparent excess populations for a wave with a phase that yields the highest apparent excess population for a given wave amplitude and wavelength. There are small but observable effects in the limiting case of no waves (background atmosphere, see Fig. 5 (a)), and the apparent excess population increases for longer wavelengths and stronger waves similarly to the non-linearity of the fit. Extreme waves with 40 K amplitude at an altitude of 90 km can cause up to 1.3 apparent excess populations in the $J'$=6.5 line.

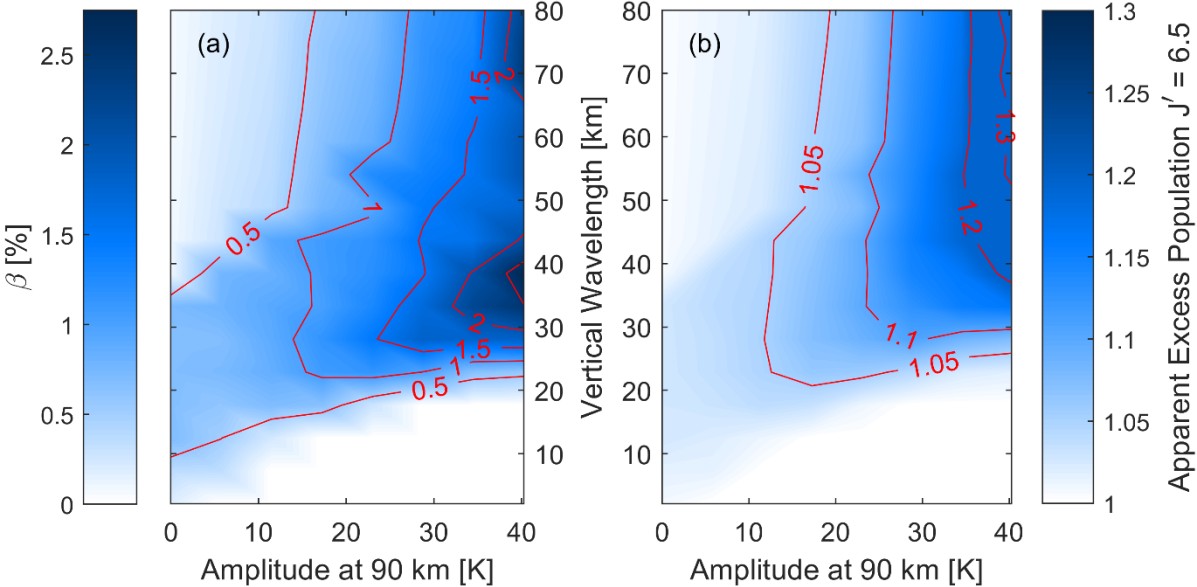

Figure 6: (a) The $\beta$-value from the non-linear fit (Equation (10)), and (b), the apparent excess population of the $J'$ = 6.5 ($N = 7$) of the (7,4) Meinel transition for the ensemble wave spectrum. The values are shown as a function of the wavelength, $\lambda_w$, and the amplitude, $A_w$. Each value represents the phase $\varphi_w$ where the apparent excess population was largest. Extreme waves (tides) can show a non-linearity of about 2% corresponding to an apparent excess population of 1.3.

While Fig. 6 shows the (7,4) transition, Fig. 7 shows the same analysis for the commonly observed (3,1) transition. Both the $\beta$-value and the apparent excess population look qualitatively similar to the (7,4) band, but the effect of waves on the (3,1) transition is about 30 % stronger. That means that the non-linearity exceeds 2.5 % and the apparent excess population of the $J' = 6.5$ level is up to 1.4 for the largest waves shown here.

Fig. 6 and 7 show the results for the phase that created the largest apparent excess population. Hence, the phase is not constant for each point in these two figures, but rather varies to show the largest apparent excess population for the wave amplitude and wavelength in question. The mean effect of all different phases simulated is independent of the transition and varies between 20 % and 40 % of the maximum effect presented in Fig. 6 and 7.

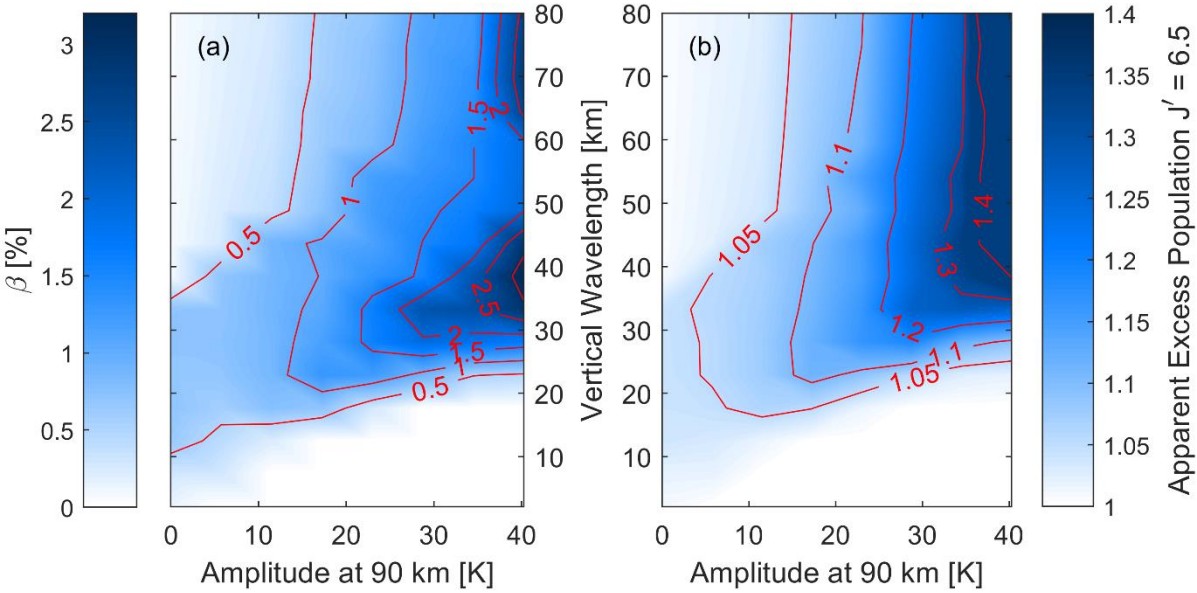

Figure 7: As Fig. 6, but this time for the (3,1) transition. Both the β-value from the non-linear fit in (a) and the apparent excess population of the $J' = 6.5$ upper level in (b) show a stronger effect than seen for the (7,4) transition in Fig. 6.

Tests showed that the difference in the apparent excess population between the (3,1) and (7,4) bands is unlikely to be due to the altitude separation of the different vibrational levels in the OH airglow layer (von Savigny et al., 2012). Repeating the analysis and weighting the $v'=3$ and 7 levels with the same VER profile yields essentially the same result, as did performing the analysis in an unperturbed isothermal background temperature profile. Instead, the difference in the apparent excess population between the (3,1) and (7,4) bands is likely due to the compressed rotational energy structure of the higher vibrational levels that lie closer to the dissociation limit. This compression of the rotational energy levels is due to the increased moment of inertia, and hence the reduced rotational constant, associated with the larger average inter-nuclear distance of the higher vibrational levels. Thus a given $J'$ level in a low vibrational state will have more rotational energy than one in a high vibrational state. Thus, for a given temperature, higher rotational levels will be thermally populated in the higher vibrational levels. These thermally populated higher rotational levels then make the perturbing effects of waves relatively less important.

Fig. 8 illustrates the dependency of the amount of apparent excess population on the different vibrational upper levels at different altitudes. For consistency, the apparent excess population of the $J' = 6.5$ upper level is again shown. All data points are for the wave presented above as an example, with an amplitude of 30 K at 90 km altitude and a vertical wavelength of 30 km.

The smallest upper vibrational level of $v' = 2$ shows an apparent excess population of about 1.28, while the highest upper vibrational level of $v' = 9$ only shows an apparent excess population of about 1.08.

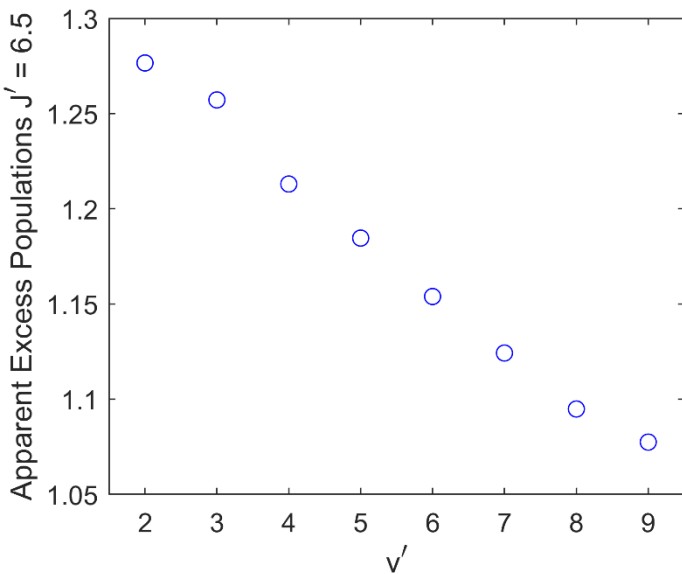

Figure 8: The maximum apparent excess population of the *J' = 6.5* upper level for a wave with wavelength 30 km and an amplitude of 30 K at 90 km altitude as a function of vibrational upper level. The effect of the waves, creating the apparent excess population decreases with rising vibrational quantum number *v'*.

## 4 Conclusions

Spectroscopic observations of the OH airglow have been commonly used to measure temperatures in the MLT. The OH radiates over an extended, Chapman-like layer that extends over several kilometres, over which the temperature is changing. Strong waves perturbing the MLT can make this change in temperature within the OH layer substantial. The simulations executed here show that these temperature profiles can create an apparent non-thermal population of the rotational levels in a given Meinel (v', v'') band. Even though the simulations calculated the rotational population distribution of the OH to be in LTE with the surrounding gas at every altitude, the integrated intensities of the higher rotational lines indicate an apparent excess population that could be misinterpreted as contributing to the NLTE effects previously reported (Cosby and Slanger, 2007; Noll et al., 2015; Pendleton et al., 1993).

We have shown in this work that the influence of a wave with an amplitude of 30 K and a vertical wavelength of 30 km at an altitude of 90 km can produce an apparent excess population of 1.12, i.e. they can explain 12 % of the effects previously ascribed to NLTE by Pendleton et al. (1993) for the (7,4) transition. Larger waves cause an apparent excess population of up to 1.3 and can therefore explain up to 30 % of these effects. . Smaller waves can also have an apparent excess population of up to 1.1, and their impact can not be ignored. Other transitions with lower vibrational quantum numbers show even higher apparent excess populations caused by this temperature-variation effect. We conclude that it is necessary to consider the temperature profile in order to infer OH rotational level population distributions from ground-based airglow observations.

**Author contribution**

CF modified the chemical model with dynamic inputs, ran the model and interpreted the model output. CF also wrote the paper. PJE supported the analysis process and paper writing. REH gave input on interpretation and paper writing.

**Competing interests**

The authors declare that they have no conflict of interest.

**Acknowledgements**

This work was supported by the Research Council of Norway/CoE under contract 223252/F50. We thank Halvor Borge, who undertook some preliminary studies on this research during his Master thesis at NTNU in the spring

semester 2018. We also thank Kate Faloon for her work on the chemical model.

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
