# Peer review of "Modelled Effects of Temperature Gradients and Waves on the"

_Atmospheric Chemistry and Physics, 2019_

## Referee Comment (RC1) · Anonymous Referee #1 · 18 Apr 2019

General comments:

This manuscript deals with an interesting effect not properly addressed by previous studies on the population of rotational levels of hydroxyl radicals in the mesosphere / lower thermosphere region of the Earth's atmosphere. The simple fact that kinetic temperature varies within the altitude range of the OH emission layer (for a wave or tidal perturbation possibly dramatically) will lead to an apparent deviation of the population of higher rotational states from LTE in ground-based observations. While this effect may not explain the majority of observed non-thermal populations of higher rotational levels, it certainly is an interesting effect that should be reported and considered in

future studies. For this reason this manuscript is of interest to the aeronomy community and should eventually be published in my opinion. I ask the authors to consider the (mainly minor) comments below.

Specific comments:

Page 3, line 15: "from the empirical model of the US Naval Research Laboratory Mass Spectrometer and Incoherent Scatter radar (NRLMSISE-00)" sounds a little odd and is misleading. I suggest something like: "from the US Naval Research Laboratory's Mass Spectrometer and Incoherent Scatter radar model (NRLMSISE-00)"

Page 4, line 1: "The temperature dependent rate coefficients for THESE reactions"

Should this perhaps be ".. for THIS reaction"? The statement seems to refer to reaction (2) only.

Page 5, line 16: "Where $A\_w(z)$ is a function of altitude"

Please provide more information on the altitude dependence of the wave amplitude, perhaps show a plot.

Page 5, same line: "and $\phi_w$ for this example is defined as zero at an altitude of 74 km"

This statement suggests that the phase is altitude dependent. Is this really the case? I assume it is constant – then the reference to a specific altitude (74 km) can be removed.

Page 5, same paragraph: It would be good to state here already, whether the assumed wave perturbation may really occur in the atmosphere, or whether this is an extreme case that essentially never occurs.

Page 6, line 12: "Where $N\_{v'}$ is the integral of the vibrational band VER"

Is this really the case? I think $N\_{v'}$ is not an integrated VER, but rather the total (integrated) population of the v' level. Otherwise the units don't fit.

Page 7, Figure 3, upper abscissa legend: Please add a space between "1.5" and "2.5"

Page 8, line 7: "the original background profile is retrieved"

I suggest replacing "retrieved" by "obtained". For me as a "retrieval person" "retrieved" here means the temperature retrieval using the OH technique.

Page 8, line 8: only a minor comment, but "higher and lower temperatures" is perhaps more accurate than "warmer and colder temperatures", because temperature cannot really be warm/cold. I leave it up to the authors to decide, whether they want to change this.

Page 9, Figure 4: It would be good to separate the two panels a little, add some space between panel a and b.

Page 9, line 18: "The first three rotational levels reproduce the same population as an atmosphere in LTE characterised by a single temperature"

It's not entirely clear, whether this statement refers to the results of Pendleton et al. (1993). I think this is the case and suggest stating this explicitly.

Page 10, line 7: "In Fig. 5(a) the non-linearity of the temperature fit increases with both wave strength and vertical wavelength"

Looking at the Figure, this statement is not generally correct, is it? There are different regimes showing different behavior, e.g., for a vertical wavelength of 10 km, beta decreases with increasing amplitude. Please rephrase this sentence.

Page 10, line 9: "Figure 5 (b) shows the phase that yields the highest . . ."

The Figure does not show the phase, but the apparent excess population for the phase with . . . Please rephrase.

Page 11, line 10: "Instead it is due to the"

It's not clear, what "it" refers to in this sentence. I assume it is "the difference in the apparent excess population" mentioned a few sentences above. Please state it explicitly.

Page 12, line 9: "The simulations executed here show that these temperature profiles can change the populations of the different rotational lines"

I have two comments on this sentence: 1) it is not surprising that temperature affects the rotational population of a vibrational state and I think this is not what you actually intend to say. The important point is that the vertical temperature variation leads to an apparent non-thermal population for an observer on the ground, right? 2) can "rotational lines" be populated? The rotational "states" or "levels" are populated and they give rise to the emission.

Typos etc.:

General: The section titles are all upper cases, which – I believe – is not the ACP standard.

Page 1, reaction (1): The "-" sign inside the parentheses may be interpreted as a minus sign and I think it is not really necessary.

Page 3, caption of Figure 1, line 1: "red- dashed" -> "red-dashed" ?

Page 4, line 3: "Loss processes include losses DUE to O" ?

---

## Referee Comment (RC2) · Anonymous Referee #2 · 22 Apr 2019

General comments:

This study focusses on the impact of kinetic temperature gradients in the OH airglow emission layer on column-integrated rotational level population distributions. As the populations of high rotational levels tend to originate from altitudes with higher kinetic temperatures than those related to low rotational levels, the rotational level population distribution of a fixed vibrational level cannot be described by a single temperature. The temperature tends to increase with rotational level, which looks like a non-LTE effect. The authors simulated different wave-perturbed temperature profiles combined with a chemical model for the OH emission layer (depending on the vibrational level)

[Figure]

to estimate the magnitude of the effect. A comparison with measured OH populations from Pendleton et al. (1993) then showed that this pseudo-non-LTE effect can significantly contribute to the apparent non-LTE deviations in the rotational level population distribution if the wave amplitudes are large.

There has not been a detailed study of the investigated effect in the literature, so far. Hence, it is justified to publish a paper on this topic, although I guess that the effect appears to be negligible in most cases. As the study does not discuss the distribution of true temperature gradients (either from satellite or lidar data), I would appreciate an extension of the analysis in order to better understand the relevance for observed rotational level population distributions like those from Pendleton et al. (1993) or more recent studies. For OH-based estimates of the kinetic temperature, rotational temperatures are usually derived from the first three or four rotational lines. Hence, it would also be interesting to know whether a noteworthy contribution of the discussed effect is possible. OH modelling results in the literature should not be affected as such models are calculated altitude-dependent. Only the interpretation of the column-integrated populations may change slightly.

Apart from the desired clarifications concerning the impact of the effect, the quality of the discussion needs to be significantly improved. In particular, due to the lack of precision in the use of scientific terms, the paper is often confusing and misleading (see detailed comments). Thus, this paper needs a major revision to be acceptable for publication in ACP.

Specific comments:

P.1, L.23: "kcal" is an old-fashioned physical unit. The SI equivalent would be "KJ".

P.1, L.29-30: The rotational temperature fits for the OH nascent populations used by Dodd et al. (1994) originate from Llewellyn & Long (1978).

P.1, L.30-31: Khomich et al. (2008) do not report rotational temperatures of about

10,000 K. They refer to such values only in the context of vibrational temperatures. Nevertheless, rotational temperatures of this magnitude have already been measured (Oliva et al. 2015, Kalogerakis et al. 2018).

P.1, L.32 - P.2, L1: In general, it cannot be stated that low rotational levels (the rotational quantum numbers could be mentioned) are in LTE. In the case of very low vibrational levels, this might be close to reality, but for higher vibrational levels, there appears to be a significant excess even for the lowest states (e.g. Noll et al. 2018). The problem is that the full thermalisation is not achieved due to an insufficient number of thermalising collisions in the especially short lifetimes of the high vibrational levels. Without the vibrational level changes by collisions and photon radiation, LTE would be possible.

P.2, L.8: There are additional relevant studies by Kalogerakis et al. published in ACP and Science Advances in 2019.

P.2, L.10: "the OH is in LTE": This statement is wrong. As the authors use an OH kinetic model involving non-LTE chemistry and radiation to derive the OH layer depending on vibrational level, LTE is only true for the rotational populations of a fixed vibrational level in the model. In the case of full LTE, most OH molecules would be in the vibrational ground state. Also note that a difference in the effective emission altitude for the different vibrational levels is a strong indicator of non-LTE effects. Larger non-LTE excesses are related to higher emission altitudes.

P.2, L.12-13: The use of "excess" is only correct if its definition is related to the lowest rotational levels. In the case of a temperature gradient of zero and an average temperature as the reference, there would also be negative population changes up to a certain rotational level.

P.2, L.30: "the OH" should be extended by "rotational level distributions" (see comment on P.2, L.10).

P.2, L.30-32: In fact, there is not a single OH layer if non-LTE effects contribute and OH

ro-vibrational states are considered separately. Each OH line originates from a different altitude range (even if the upper vibrational level is kept fixed) depending on the non-LTE contribution. Hence, in reality, it will be complicated to separate the temperature gradient effects from the true non-LTE effects.

P.2, L.32-33: "the emission from high rotational lines that occurs in the warm regions": This statement should be softened since a negligible contribution of the cold regions to the high rotational states would require large temperature gradients. In this respect, it would be helpful if such a criterion could be quantified in the paper.

P.3, L.8: "OH is in LTE" (see comment on P.2, L.10).

P.3, L.11: "excess population" (see comment on P.2, L.12-13).

P.4, L.5-7: The use of hot nascent OH populations stringently leads to non-LTE level populations. Therefore, it should be made clear that these calculations are only used to derive the vertical OH emission distribution for each vibrational level (which neglects the spread of emission heights depending on the rotational level). The rotational level population distributions are arbitrarily set to LTE.

P.4, L.8: The most frequent constituent of the Earth's atmosphere, $N_2$, is not considered for the collisional loss. Why?

P.4, L.10: Why is the primitive "sudden death" approach used for all kinds of collisions? This is not state of the art. There are various examples of more sophisticated models in the literature. For example, the model by Adler-Golden (1997) as well as the fast Sharma et al. (2015) process for OH vibrational relaxation by atomic oxygen collisions had a big impact.

P.4, L.11: Radiation is also an important source of non-LTE effects, especially if it is not reabsorbed nearby as in the case of OH. This is also contributes to the fact that the model in Sect. 2.1 is not able to provide OH in LTE. Hence, one should make clear that there is no consistent OH model that delivers thermalised populations. The

investigation of the temperature gradient effect therefore requires the arbitrary manipulation of population distributions. I do not criticise this approach but it should be better communicated.

P.4, L.11: "z" is not defined.

P.4, L.12: Mies (1974) does not give lifetimes. He provides Einstein-A coefficients. The set of coefficients is already quite old. Better data would be available.

P.4, L.13: Mies (1974) does not introduce omega_v'v".

P.4, L.26-29: Eq. 5 needs to be revised for several reasons. N_J' can only be a relative population as an absolute population is already given by N_v' if the descriptions from Sect. 2.1 still hold. A better solution would be to use N_v'J' from Eq. 6. As omega_v'v"/tau_v' corresponds to A_v'v", the equation shows a product of Einstein-A coefficients, which looks wrong. It would be OK to replace both terms by A_v'v"J'J". Moreover, V_v'v"J'J" and N_J' are also functions of z. Finally, level splitting by spin-orbit coupling is not considered (symbol: i or F). This should be included as Fig. 2 proves that it was considered for the model.

P.4, L.29 - P.5, L.2: Eq. 6 also needs to be revised since the partition function Q_v'(T_rot) (Mies 1974) is missing. Otherwise the rotational level population distribution is not correctly normalised. It would be better to replace E_J' by E_v'J' as the level energy also depends on v'. Moreover, the electronic substates are not provided. Finally, the z dependences of N_v'J' and N_v' are not given.

P.5, L.4: "the OH is in LTE" (see comment on P.2, L.10).

P.5, L.7-8: The statement related to the cool and warm regions should be weakened since a complete separation of the rotational level populations cannot be expected for typical temperature gradients.

P.5, L.11: "assuming LTE" should be extended by "for the rotational level populations".

P.6, L.2: "ergs cmˆ-3 sˆ1" are quite old-fashioned units. Moreover, "sˆ-1" would be correct.

P.6, L.7-8: The statement is only correct for the plotted phase of the wave. A phase shift of 180 deg would change the effective sign of the temperature gradient in the OH layer.

P.6, L.10-13: In the same way as Eqs. 5 and 6, Eq. 8 needs to be improved. I propose to write I_v'v"J'J", A_v'v"J'J", E_v'J', and E_v'. This is further modified if the electronic substates are introduced (e.g. by adding i' or F'). Finally, Q_v'(T_rot) is also missing here.

P.6, L.13-16: As Eq. 9 is very similar to Eq. 8, the same proposals are relevant. Depending on the changes, the symbols in Fig. 3 need to be adapted.

P.7, L.1-2: "the lowest three rotational levels": This obviously refers to the F = 1 sub-states.

P.7, L.10: The populations are related to P_1-branch lines.

P.7, L.16 - P.8, L.1: This is again an equation which should be improved (see comments on previous equations).

P.8, L.1-2: It is stated that the beta parameter was used to improve the fit of the population distribution in Fig. 3. However, the actual value of beta is not provided.

P.8, L.28-30: beta is also not given for the scenarios in Fig. 4.

P.8, L.32-34: "the distribution of integrated rotational line intensities" for the fitting procedure does not appear to be limited in terms of the rotational levels. If high rotational levels contribute to the fit, the low rotational levels will show underpopulations. In this case, "excess" will be a wrong term.

P.9, L.10: Excess populations for P_1-branch lines are shown. The excess is probably related to the first three rotational levels. This information should be given in the

caption.

P.9, L.14-26: The comparison with the measurements of Pendleton et al. (1993) would be less awkward if the population ratios from their Fig. 16 are also plotted in Fig. 4. It should not be a problem to add histograms of different colour or transparency to the two subfigures.

P.9, L.22: This is actually the only place where "P_1" is given. This notation will be understood if the related cases are consistently changed in the paper.

P.9, L.26 - P.10, L.2: This conclusion sounds more dramatic than it is since OH kinetic models are usually resolved in altitude, i.e. the models implicitly include the temperature gradient effect. Hence, comparisons with ground-based measurements should be trustworthy without any change in the modelling approach.

P.10, L.5: It is not clear which lines were used for the baseline fit for the derivation of the excess population. It might be helpful to define the term "excess population" at the first occasion and then to state that this definition is used for the rest of the paper.

P.10, L.6-7: "The figure shows the (7,4) transition" sounds a bit strange. The figure shows results related to the (7,4) band. There are other similar sentences in the paper. As I would primarily expect that "transition" describes a spectral line, "band" or "vibrational transition" appear to be clearer.

P.10, L.9-10: The discussion in Sect. 3 only focusses on the "highest apparent excess population" for waves of a given amplitude and vertical wavelength. It would also be helpful to briefly quantify the average effect and the related variation. As already discussed in the context of the general comments, it is also important to know what the real distribution of temperature gradients for the OH layer is (or equivalently: the true frequency of the different simulated waves) in order to estimate the real impact of the temperature gradient effect. These results should be compared to the true non-LTE effects (as for the two examples in Fig. 4).

[Figure]

P.12, L.3: Just writing "The apparent excess population" is incomplete. In fact, it is "The maximum apparent excess population" for the described wave.

P.12, L.10: "the OH in LTE" (see comment on P.2, L.10).

P.12, L.18: It depends on the research goal and the related significance of the effect whether the consideration of the effect is really "necessary". Concerning the estimate of the true impact, the paper should be improved.

P.12, L.18: "the rotational distribution of the OH" might be replaced by "OH rotational level population distributions", which sounds better.

Technical corrections:

Whole paper: I have identified four different styles for the vibrational level symbol "v" (see e.g. P.5). In several equations, it is even an upper case letter. This should be harmonised. Other symbols should be checked in the same way.

P.9, L.2: "rad" should not be in italics.

---

## Author Comment (AC1) · 27 Aug 2019

Answer to Anonymous Referee #1

General comments:

This manuscript deals with an interesting effect not properly addressed by previous studies on the population of rotational levels of hydroxyl radicals in the mesosphere / lower thermosphere region of the Earth's atmosphere. The simple fact that kinetic temperature varies within the altitude range of the OH emission layer (for a wave or tidal perturbation possibly dramatically) will lead to an apparent deviation of the population of higher rotational states from LTE in ground-based observations. While this effect may not explain the majority of observed non-thermal populations of higher rotational levels, it certainly is an interesting effect that should be reported and considered in future studies. For this reason this manuscript is of interest to the aeronomy community and should eventually be published in my opinion. I ask the authors to consider the (mainly minor) comments below.

*We thank the referee for all comments and helpful feedback. We address each comment in turn below.*

Specific comments:

Page 3, line 15: "from the empirical model of the US Naval Research Laboratory Mass Spectrometer and Incoherent Scatter radar (NRLMSISE-00)" sounds a little odd and is misleading. I suggest something like: "from the US Naval Research Laboratory's Mass Spectrometer and Incoherent Scatter radar model (NRLMSISE-00)"

*The text has been changed as suggested.*

Page 4, line 1: "The temperature dependent rate coefficients for THESE reactions" Should this perhaps be ".. for THIS reaction"? The statement seems to refer to reaction (2) only.

*The text has been changed as suggested.*

Page 5, line 16: "Where A_w(z) is a function of altitude" Please provide more information on the altitude dependence of the wave amplitude, perhaps show a plot.

*We follow here the wave breaking mechanism of (Holton, 1982). To clarify the amplitude behaviour with altitude, we have added the following passage and figure to the text:*

"Where $\varphi_w$ is the wave's phase, and $A_w(z)$ is a function of altitude so as not to exceed the dry adiabatic lapse rate. The lower edge of the model is 74 km altitude. The wave amplitude as a function of altitude is shown in Figure 2 for the case of a wave with an amplitude of 10 K at 74 km altitude and a vertical wavelength of 15 km. This example is given for an isothermal atmosphere of 200 K (Figure 2 (a), dashed-black). The wave grows in amplitude with altitude to conserve energy (dashed-blue), but at regions where the lapse rate exceeds the dry adiabatic lapse rate (here between 90 and 95 km, and between 110 and 115 km), the wave loses energy and the amplitude decreases (Holton, 1982). The breaking wave is shown in red in Figure 2 (a). Figure 2 (b) shows the instantaneous lapse rate (change in temperature with altitude) of the non-breaking wave (dashed-blue) and the breaking wave (red), which never crosses the dry adiabatic lapse rate of -10 K/km (dashed-black). Figure 2 (c) shows the amplitude of the non-breaking wave (dashed-blue) which increases exponentially and for the breaking wave (red), which decreases at the altitudes where the wave dissipates energy.

[Figure]

Figure 2: (a) Example of a wave of amplitude 10 K at 74 km altitude perturbing an isothermal atmosphere at T = 200 K (dotted-black). The dashed-blue line represents the wave without any breaking, and the red line is the breaking wave as used in the model. (b) The rate of change of temperature with altitude for the wave shown in (a). When the temperature changes faster than -10 K/km (dashed-black) the wave breaks. (c) At these altitudes, the amplitude of the breaking wave (red) decreases, while the non-breaking wave continues to grow exponentially (dashed-blue)."

Page 5, same line: "and $nphi\_w$ for this example is defined as zero at an altitude of 74 km" This statement suggests that the phase is altitude dependent. Is this really the case? I assume it is constant – then the reference to a specific altitude (74 km) can be removed.

*The phase shift $\phi_w$ is not altitude dependent, as the referee said. The text has been changed to be less confusing. It now reads:*

"Where $\varphi_w$ is the wave's phase (constant with altitude), and $A_w(z)$ is a function of altitude so as not to exceed the dry adiabatic lapse rate. The lower edge of the model is 74 km altitude."

Page 5, same paragraph: It would be good to state here already, whether the assumed wave perturbation may really occur in the atmosphere, or whether this is an extreme case that essentially never occurs.

*We agree that is it good to state this already here. We have added the following sentence to the text.*

"Waves with similar amplitudes have been observed at these altitudes (Picard et al., 2004)".

Page 6, line 12: "Where N_v' is the integral of the vibrational band VER" Is this really the case? I think N_v' is not an integrated VER, but rather the total (integrated) population of the v' level. Otherwise the units don't fit.

*N_v' is indeed the integral over the population, or in terms of VER, you can also put the integral as $N_{v'}(z) \cdot \omega_{v'v''} / \tau_{v'}$. We clarified this in the text. It now reads:*

"Where $N_{v'}$ is the integral of the vibrational population $N_{v'}(z)$ over altitude, and $T$ is the effective rotational temperature of the altitude-integrated spectrum."

Page 7, Figure 3, upper abscissa legend: Please add a space between "1.5" and "2.5"

*We have rotated the labels by 90 degrees to increase readability. The new figure now looks like this:*

[Figure]

"

Page 8, line 7: "the original background profile is retrieved" I suggest replacing "retrieved" by "obtained". For me as a "retrieval person" "retrieved" here means the temperature retrieval using the OH technique."

*The text has been changed as suggested.*

Page 8, line 8: only a minor comment, but "higher and lower temperatures" is perhaps more accurate than "warmer and colder temperatures", because temperature cannot really be warm/cold. I leave it up to the authors to decide, whether they want to change this.

*The text has been changed as suggested.*

Page 9, Figure 4: It would be good to separate the two panels a little, add some space between panel a and b.

*We have added some space between these two panels. The figure now looks like this:*

"

[Figure]

[Figure]

Figure 5: The calculated apparent excess population of the OH (7,4) P($J'$) lines relative to the Boltzmann population of the fitted temperature to the lowest three lines. Red on the left is calculated with the climatological temperature gradient shown from NRLMSISE-00 above Boulder, Colorado (40.0° N; 105.6° W) in mid-July. Blue on the right is calculated with a wave of $A_w = 30$ K at 90 km, $\lambda_w = 30$ km and a phase that yields the maximum effect, $\varphi_w = 4.9$ rad. This is the same wave as presented in Figure 4. The distribution has a β = 1.9 %. The grey bars represent the measurements ascribed to NLTE effects from Pendleton et al. (1993)."

Page 9, line 18: "The first three rotational levels reproduce the same population as an atmosphere in LTE characterised by a single temperature" It's not entirely clear, whether this statement refers to the results of Pendleton et al. (1993). I think this is the case and suggest stating this explicitly.

*This statement refers to both this paper and the one of Pendleton et al. (1993). We have changed this sentence to clarify this. It now reads:*

"The first three rotational levels in this study and in Pendleton et al. (1993) have populations that follow a Boltzmann distribution characterized by a single temperature, and above that, there is excess population."

Page 10, line 7: "In Fig. 5(a) the non-linearity of the temperature fit increases with both wave strength and vertical wavelength" Looking at the Figure, this statement is not generally correct, is it? There are different regimes showing different behavior, e.g., for a vertical wavelength of 10 km, beta decreases with increasing amplitude. Please rephrase this sentence.

*The referee is correct. We have clarified this in the text. It now reads:*

"Figure 6 (a) shows that for vertical wavelengths above about 20 km the non-linearity of the temperature fit increases with both wave strength and vertical wavelength approaching 2% of the linear temperature variation (see Equation 10). For shorter vertical wavelengths, the non-linearity is smaller (at about 0.5 %) and decreases with increasing wave amplitude."

Page 10, line 9: "Figure 5 (b) shows the phase that yields the highest : : :" The Figure does not show the phase, but the apparent excess population for the phase with : : : Please rephrase.

*The figure shows the apparent excess population for a wave with a given wavelength and amplitude using the phase that yields the greatest effect. We have now clarified this in the text, which reads:*

"Figure 6 (b) shows the apparent excess populations for a wave with a phase that yields the highest apparent excess population for a given wave amplitude and wavelength."

Page 11, line 10: "Instead it is due to the" It's not clear, what "it" refers to in this sentence. I assume it is "the difference in the apparent excess population" mentioned a few sentences above. Please state it explicitly.

*The "it" refers to the difference in the apparent excess population between the (3,1) and (7,4) bands. This has been clarified and the sentence now reads:*

"Instead, the difference in the apparent excess population between the (3,1) and (7,4) bands is likely to be due to the compressed rotational energy structure of the higher vibrational levels that lie closer to the dissociation limit."

Page 12, line 9: "The simulations executed here show that these temperature profiles can change the populations of the different rotational lines" I have two comments on this sentence: 1) it is not surprising that temperature affects the rotational population of a vibrational state and I think this is not what you actually intend to say. The important point is that the vertical temperature variation leads to an apparent non-thermal population for an observer on the ground, right? 2) can "rotational lines" be populated? The rotational "states" or "levels" are populated and they give rise to the emission.

*We agree with the referee on both points. We have changed the sentence to clarify the apparent non-thermal population and exchanged "line" with "level". The whole sentence now reads:*

"The simulations executed here show that these temperature profiles can create an apparent non-thermal population of the rotational levels in a given Meinel (v', v'') band."

Typos etc.:

General: The section titles are all upper cases, which – I believe – is not the ACP standard.

*The section titles have been changed as suggested.*

Page 1, reaction (1): The "-" sign inside the parentheses may be interpreted as a minus sign and I think it is not really necessary.

*The minus sign was supposed to indicate that this energy is the excitation of the $OH^*$. We agree however that it might be confusing and have deleted this sign.*

Page 3, caption of Figure 1, line 1: "red- dashed" -> "red-dashed" ?

*The text has been changed as suggested*

Page 4, line 3: "Loss processes include losses DUE to O" ?

*The text has been changed as suggested*

References:

Holton, J. R.: The Role of Gravity Wave Induced Drag and Diffusion in the Momentum Budget of the Mesosphere, J Atmos Sci, 39, 791-799, 1982.
Pendleton, W. R., Espy, P. J., and Hammond, M. R.: Evidence for non-local-thermodynamic-equilibrium rotation in the OH nightglow, J Geophys Res-Space, 98, 11567-11579, 1993.

Picard, R. H., Wintersteiner, P. P., Winick, J. R., Mertens, C. J., Mlynczak, M. G., III, J. M. R., Gordley, L. L., Ward, W. E., She, C. Y., and O'Neil, R. R.: Tidal and layer structure in the mesosphere and lower thermosphere from TIMED/SABER $CO_2$ 15-μm emission, SPIE, 2004.

---

## Author Comment (AC2) · 27 Aug 2019

Answer to Anonymous Referee #2

General comments:

This study focusses on the impact of kinetic temperature gradients in the OH airglow emission layer on column-integrated rotational level population distributions. As the populations of high rotational levels tend to originate from altitudes with higher kinetic temperatures than those related to low rotational levels, the rotational level population distribution of a fixed vibrational level cannot be described by a single temperature. The temperature tends to increase with rotational level, which looks like a non-LTE effect. The authors simulated different wave-perturbed temperature profiles combined with a chemical model for the OH emission layer (depending on the vibrational level) to estimate the magnitude of the effect. A comparison with measured OH populations from Pendleton et al. (1993) then showed that this pseudo-non-LTE effect can significantly contribute to the apparent non-LTE deviations in the rotational level population distribution if the wave amplitudes are large.

There has not been a detailed study of the investigated effect in the literature, so far. Hence, it is justified to publish a paper on this topic, although I guess that the effect appears to be negligible in most cases. As the study does not discuss the distribution of true temperature gradients (either from satellite or lidar data), I would appreciate an extension of the analysis in order to better understand the relevance for observed rotational level population distributions like those from Pendleton et al. (1993) or more recent studies. For OH-based estimates of the kinetic temperature, rotational temperatures are usually derived from the first three or four rotational lines. Hence, it would also be interesting to know whether a noteworthy contribution of the discussed effect is possible. OH modelling results in the literature should not be affected as such models are calculated altitude-dependent. Only the interpretation of the column-integrated populations may change slightly.

Apart from the desired clarifications concerning the impact of the effect, the quality of the discussion needs to be significantly improved. In particular, due to the lack of precision in the use of scientific terms, the paper is often confusing and misleading (see detailed comments). Thus, this paper needs a major revision to be acceptable for publication in ACP.

*We thank the referee for their extensive and helpful comments.*

*This study uses a temperature profile from the reanalysis NRLMSIS-00, which is based on "realistic temperature gradients". We have also been careful to ensure that the waves we present have been reported in the literature using a variety of observational techniques and are therefore physical (Hagan et al., 1999; Oberheide et al., 2011; Picard et al., 2004; She et al., 2002; Shepherd and Fricke-Begemann, 2004; Zhang et al., 2006). The results presented here describe the temperature gradients that arise over the range of these observed waves. We believe that a systematic scan of different waves spanning the range of temperature gradients arising from observed waves presents a stronger argument than a calculation of a specific observation from satellite or lidar, each of which has specific observational bias.*

*The referee is right that OH temperatures are typically derived from lines with N<5, which are the first three (for the P-branch) or four (for the Q- and R-branches) lines. These lines are not affected, or are only marginally affected, by the findings presented in this paper as is clearly stated in the manuscript. The focus of this paper is not to argue an impact on the majority of OH temperature*

*calculations, but rather to highlight the impact on attempts to quantify other contributions to the excess populations observed in the higher lying rotational levels.*

*We have responded to all comments in order below and hope that the scientific precision is now acceptable to the referee and that we have cleared up any confusing or misleading text. Again, we thank the referee for their comments and for their help in improving the quality of the paper.*

Specific comments:

P.1, L.23: "kcal" is an old-fashioned physical unit. The SI equivalent would be "KJ".

*We deleted the kcal unit and use kJ now instead. The whole passage now reads:*

"The 8-km thick airglow layer is created at about 90 km altitude (Baker and Stair, 1988; Xu et al., 2012) by the highly exothermic reduction of ozone:

$$H + O_3 \rightarrow OH^* + O_2 \ (5.3 \cdot 10^{-19} J) \tag{1}$$

The excess heat of reaction, ~$5.3 \cdot 10^{-19}$J, produces the $OH^*$ in excited vibrational quantum levels of *v'*=6-9 (e.g. Mlynczak and Solomon (1993))."

P.1, L.29-30: The rotational temperature fits for the OH nascent populations used by Dodd et al. (1994) originate from Llewellyn & Long (1978).

*The v' = 9 temperature is indeed from Llewellyn % Long (1978), while the v' = 6 temperature is from Dodd et al. (1994). This has been corrected. The text now reads:*

"Llewellyn & Long (1978) reported a nascent temperature of 760 K for v'= 9, whereas others report temperatures as high as 9000-10000 K (Dodd et al, 1993; Oliva et al., 2015; Kalogerakis et al., 2018)."

P.1, L.30-31: Khomich et al. (2008) do not report rotational temperatures of about 10,000 K. They refer to such values only in the context of vibrational temperatures. Nevertheless, rotational temperatures of this magnitude have already been measured (Oliva et al. 2015, Kalogerakis et al. 2018).

*We have changed the references here as suggested.*

P.1, L.32 - P.2, L1: In general, it cannot be stated that low rotational levels (the rotational quantum numbers could be mentioned) are in LTE. In the case of very low vibrational levels, this might be close to reality, but for higher vibrational levels, there appears to be a significant excess even for the lowest states (e.g. Noll et al. 2018). The problem is that the full thermalisation is not achieved due to an insufficient number of thermalizing collisions in the especially short lifetimes of the high vibrational levels. Without the vibrational level changes by collisions and photon radiation, LTE would be possible.

*We use the term LTE loosely in the sense that the distribution of radiation from the levels can be describes by a single temperature Boltzmann distribution. For low rotational levels N ≤ 4, there is evidence that this temperature is consistent with the atmospheric temperature (Oberheide et al., 2006; von Zahn et al., 1987). As such, we have taken the approach that at each altitude, the temperature characterizing the Boltzmann distribution is the local atmospheric temperature, and examine only the deviations to that temperature caused by integrating each rotational line intensity through the temperature gradients across the layer.*

*We have made this passage clearer, adding specifically which levels we are discussing and that the studies referenced find a single temperature describes the Boltzmann distribution. We have also added a reference to the paper of Noll et al. at the end of this paragraph. Since our study however uses strict LTE conditions in the model and has no NLTE effects programmed in the model, this paragraph remains quite short, since its intention is to give an overview of the general research done in the field rather than to discuss or make inferences on the impact of NLTE effects on the lower rotational levels.*

*The new paragraph now reads:*

"Low rotational levels (with N ≤ 4) with energy separations less than $kT$, the amount typically exchanged during collisions, have been observed to have efficient energy transfer in the thermalisation process (Maylotte et al., 1972; Polanyi and Sloan, 1975; Polanyi and Woodall, 1972). Thus, emission from these states has been observed to be characterised by a single temperature Boltzmann distribution (Harrison et al., 1971; Harrison et al., 1970; Pendleton et al., 1993; Perminov et al., 2007; Sivjee et al., 1972; Sivjee and Hamwey, 1987). However, emission observed from the higher rotational levels (N>4), where the energy separation exceeds $kT$, has indicated an anomalous, non-thermalized population that cannot be described using the same Boltzmann temperature that characterizes the lower rotational levels (Cosby and Slanger, 2007; Dodd et al., 1994; Kalogerakis, 2019; Noll et al., 2015; Pendleton et al., 1989; Pendleton et al., 1993). In keeping with the terminology employed by Pendleton et al. (1993), these two conditions are referred to as Local Thermodynamic Equilibrium (LTE) and Non-Local Thermodynamic Equilibrium (NLTE), respectively. The term LTE as used here is not technically correct as it does not account for radiative effects. Instead it relates only to the collisional distribution of the rotational levels being characterized by the temperature of the surrounding gas, as has been done in Pendleton et al. (1993). Work is currently underway to use observations of the excess populations in the high rotational levels of the OH airglow to determine state-to-state quenching coefficients and to understand the thermalisation process in OH (Kalogerakis et al., 2018)."

P.2, L.8: There are additional relevant studies by Kalogerakis et al. published in ACP and Science Advances in 2019.

*See new text in the previous comment.*

P.2, L.10: "the OH is in LTE": This statement is wrong. As the authors use an OH kinetic model involving non-LTE chemistry and radiation to derive the OH layer depending on vibrational level, LTE is only true for the rotational populations of a fixed vibrational level in the model. In the case of full LTE, most OH molecules would be in the vibrational ground state. Also note that a difference in the effective emission altitude for the different vibrational levels is a strong indicator of non-LTE effects. Larger non-LTE excesses are related to higher emission altitudes.

*In addition to the responses to P1, L32 and P2, L1 we have changed this phrase to "the rotational population distribution of the OH is in LTE" for clarification. The whole paragraph now reads:*

"Here we examine the effects of temperature gradients in the OH emission region on the resulting vertically integrated spectrum of the Meinel $\Delta v=2$ sequence. To achieve this, model work was executed, where the model assumes that for each vibrational level, the rotational population distribution of the OH is in LTE at every altitude. The emission in each rotational line is then integrated vertically."

P.2, L.12-13: The use of "excess" is only correct if its definition is related to the lowest rotational levels. In the case of a temperature gradient of zero and an average temperature as the reference, there would also be negative population changes up to a certain rotational level.

*We defined the word excess here as relative to the expected Boltzmann population due to a single temperature, which is characterised by the lines with N≤4. However, because these negative excess populations all are <1% and very small compared to the effects of higher lines, we decided to keep the term. We agree however that the term should be better defined here (see also comment below) and we have added a further sentence here. The context now reads:*

"We find that even if the OH rotational levels are in strict LTE with the surrounding atmosphere, the temperature gradients through the OH emission region will create apparent excess emission in the higher OH rotational lines. Here we calculate the apparent excess population relative to the Boltzmann population expected using the temperature determined by the population of rotational levels with N ≤ 4. This excess population can be incorrectly interpreted as due to NLTE effects, affecting the subsequent calculations of the thermalisation process."

P.2, L.30: "the OH" should be extended by "rotational level distributions" (see comment on P.2, L.10).

*We have changed the text accordingly. The whole sentence now reads:*

"Thus the OH rotational level distributions, even if thermalised at each altitude, will be described by different temperatures at each of those altitudes."

P.2, L.30-32: In fact, there is not a single OH layer if non-LTE effects contribute and OH ro-vibrational states are considered separately. Each OH line originates from a different altitude range (even if the upper vibrational level is kept fixed) depending on the non- LTE contribution. Hence, in reality, it will be complicated to separate the temperature gradient effects from the true non-LTE effects.

*The referee is of course right in saying that non-LTE effects will make the situation more complex. The aim of this paper is to demonstrate that even without NLTE effects there is an important impact on the rotational distribution of the OH airglow. We agree that it will be complicated to separate the temperature gradient effects from the true non-LTE effects. This paper makes no attempt to explain the observed NLTE effects. However, even with NLTE effects, the sentence in question here would still be true: "Any instrument that integrates through the OH layer will therefore not see rotational line emission resulting from a single, average temperature, but from the whole span of temperatures present in the layer." We do not see that this statement is in conflict with the reviewer's comment, and prefer to keep the sentence as is, but thank the referee for the further explanation.*

P.2, L.32-33: "the emission from high rotational lines that occurs in the warm regions": This statement should be softened since a negligible contribution of the cold regions to the high rotational states would require large temperature gradients. In this respect, it would be helpful if such a criterion could be quantified in the paper.

*We have added a citation to the paper of Picard et al. (2004), who measured temperature changes of 60 between 85 and 95 km altitude. Additionally, the reported wave amplitudes in the mesopause region are consistent with temperature gradients up to this order. Since such large perturbations due to waves/tides are possible and observed, and even smaller gradients are shown in our results to have a significant impact, this sentence would appear to be accurate.*

P.3, L.8: "OH is in LTE" (see comment on P.2, L.10).

*See also answer to the comment P.2, L.10. We have also made the wording here more precise. The sentence now reads:*

"We utilise a steady-state model of the OH VER, described below, to synthesise individual synthetic rotational spectra at 1 km intervals from 74 to 110 km. The model assumes that for each vibrational level, the rotational population distribution of the OH is in LTE (i.e. a single-temperature Boltzmann distribution) with the local temperature at each altitude."

P.3, L.11: "excess population" (see comment on P.2, L.12-13).

*See also answer to comment P2, L12-13. After already defining the term more clearly, we feel confident in keeping this text.*

P.4, L.5-7: The use of hot nascent OH populations stringently leads to non-LTE level populations. Therefore, it should be made clear that these calculations are only used to derive the vertical OH emission distribution for each vibrational level (which neglects the spread of emission heights depending on the rotational level). The rotational level population distributions are arbitrarily set to LTE.

*We have specified more clearly that the rotational population calculations are performed for each vibrational level. The text now reads:*

"was also used to calculate the production rate of OH$^*$ for each vibrational level, using the reaction rate coefficient (Sander et al., 2003). Due to the exothermicity of reaction 4, vibrational levels from v'=6 to 9 can be populated. The production of each vibrational level OH$^*$(v') is calculated using the branching ratios from Sander et al. (2019)."

P.4, L.8: The most frequent constituent of the Earth's atmosphere, $N_2$, is not considered for the collisional loss. Why?

*The rate coefficient for OH quenching with $N_2$ is small, and Knutsen et al. (1996) were only able to provide an upper limit. Since the $O_2/N_2$ mixing ratio is virtually constant up to the turbopause, the $O_2$ would be 3 times more effective at quenching than the $N_2$. Thus, neglecting the $N_2$ would be well within the uncertainty of the $O_2$ rate coefficient and would not significantly affect the altitude distribution of the OH.*

P.4, L.10: Why is the primitive "sudden death" approach used for all kinds of collisions? This is not state of the art. There are various examples of more sophisticated models in the literature. For example, the model by Adler-Golden (1997) as well as the fast Sharma et al. (2015) process for OH vibrational relaxation by atomic oxygen collisions had a big impact.

*We thank the referee for pointing this out. The more sophisticated quenching models primarily change the relative vibrational distributions. However, the altitude profile of a single vibrational level is relatively unchanged, which is the primary consideration in this work. We have compared the vibrational distribution of single vibrational levels generated here with those of Adler-Golden (1997) and found them to be in good agreement. A stronger effect on the altitude profile is the profile of atomic oxygen, which is known from the SABER measurements to be highly variable. However, for MSIS atomic oxygen distributions, we are confident that our simple model reproduces the altitude distribution of a single vibrational level adequately for the purposes of this study. A small addendum was added to the paragraph mentioning the comparison with the Adler-Golden results.*

*However, a mistake was made in the description of this model. The sudden death criterion is used for the O quenching, while $O_2$ and $CO_2$ quench for steps of one vibrational unit. These are the major quenching channels for these species (Chalamala and Copeland, 1993; Dodd et al., 1991; Dyer et al., 1997; Knutsen et al., 1996; McDade and Llewellyn, 1987). We have changed the text here to correct this mistake. It now reads:*

"Collisional loss for each $OH^*(v')$ vibrational level was calculated for collisions with O, $O_2$ and $CO_2$ using the rate coefficients of Dodd et al. (1991), Knutsen et al. (1996), Dyer et al. (1997) and Chalamala and Copeland (1993). The model assumes quenching to the ground vibrational state, known as "sudden death" for the O, and stepwise quenching by one vibrational unit for the $O_2$ and the $CO_2$ (McDade and Llewellyn, 1987). The relative shape and peak height of the altitude profiles of the individual vibrational levels agree closely with those of the more sophisticated model of Adler-Golden (1997)"

P.4, L.11: Radiation is also an important source of non-LTE effects, especially if it is not reabsorbed nearby as in the case of OH. This is also contributes to the fact that the model in Sect. 2.1 is not able to provide OH in LTE. Hence, one should make clear that there is no consistent OH model that delivers thermalised populations. The investigation of the temperature gradient effect therefore requires the arbitrary manipulation of population distributions. I do not criticise this approach but it should be better communicated.

*We now clarify the "misuse" of the terms LTE and non-LTE when we first introduce them in the body of the paper. Please see also the answer to the comment on P1, L32 – P2, L1.*

"The term LTE as used here is not technically correct as it does not account for radiative effects. Instead it relates only to the collisional distribution of the rotational levels being characterized by the temperature of the surrounding gas, as has been done in Pendleton et al. (1993). "

P.4, L.11: "z" is not defined.

*We have added the definition of z as the altitude. The text now reads:*

"The total radiative loss from each vibrational level is given by $N_{v'}(z) / \tau_{v'}$, where $N_{v'}(z)$ is the concentration of the hydroxyl $v'$ vibrational level at the altitude z and its lifetime, $\tau_{v'}$, are calculated from Mies (1974)."

P.4, L.12: Mies (1974) does not give lifetimes. He provides Einstein-A coefficients. The set of coefficients is already quite old. Better data would be available.

*We thank the referee for this comment. It is true that Mies calculated Einstein coefficients rather than lifetimes. We used these to calculate the lifetimes ourselves. This is now written more precisely in the paper. (See new text in the previous comment.) However, while the distribution of line strengths within the rotational levels are different with different authors, primarily due to vibration-rotation coupling, these differences tend to average out. This is because the differences are in different directions for the R- and P- branches, and virtually zero for the Q-Branch.  Thus the lifetimes of the vibrational levels (the inverse of the sum of the individual line strengths) is relatively unchanged.*

P.4, L.13: Mies (1974) does not introduce omega_v'v".

*Mies (1974) does, however, give all the important Einstein coefficients to  calculate the branching ratio as the ratio of the sum of A-coefficients for one level divided by the total sum of all A-coefficients. We have specified this in the text, which now reads:*

"The total VER of any $v'$ to $v''$ vibrational transition is then given by $V_{v'v''}(z) = \omega_{v'v''} \cdot N_{v'}(z) / \tau_{v'}$, where $\omega_{v'v''}$ is the vibrational branching ratio calculated from the Einstein coefficients from Mies (1974)."

P.4, L.26-29: Eq. 5 needs to be revised for several reasons. N_J' can only be a relative population as an absolute population is already given by N_v' if the descriptions from Sect. 2.1 still hold. A better solution would be to use N_v'J' from Eq. 6. As omega_v'v"/tau_v' corresponds to A_v'v", the equation shows a product of Einstein-A coefficients, which looks wrong. It would be OK to replace both terms by A_v'v"J'J". Moreover, V_v'v"J'J" and N_J' are also functions of z. Finally, level splitting by spinorbit coupling is not considered (symbol: i or F). This should be included as Fig. 2 proves that it was considered for the model.

*We have changed the equation according to the suggestions of the referee. We have also now explained that we only look specifically at the 3/2 electronic subset and drop all further references to the F quantum number for readability. The new passage now reads:*

"
$$V_{v'v''J'J''}(z) = \frac{\omega_{v'v''}}{\tau_{v'}} N_{v',J'}(z) \cdot A_{v',v'',J'J''},$$
(2)

where $A_{J'J''}$ is the Einstein coefficient for the rotational transition and $N_{v',J'}$ is the population of the upper rotational level, *J'*. We concentrate in this paper on the 3/2 electronic subset of the OH airglow and drop the spin-orbit splitting quantum number F in all equations for readability."

P.4, L.29 - P.5, L.2: Eq. 6 also needs to be revised since the partition function Q_v'(T_rot) (Mies 1974) is missing. Otherwise the rotational level population distribution is not correctly normalised. It would be better to replace E_J' by E_v'J' as the level energy also depends on v'. Moreover, the electronic substates are not provided. Finally, the z dependences of N_v'J' and N_v' are not given.

*We have changed the equation according to the suggestions from the referee. The section now reads:*

"
$$N_{v',J'}(z) = \frac{N_{v'}(z)2(2J'+1)}{Q_{v'}(T(z))} \cdot exp\left(\frac{-E_{v',J'}}{k \cdot T(z)}\right)$$
(3)

Where $N_{v'}$ is the total population of the *v'* vibrational level from the model, $E_{v',J'}$ is the energy of the v',J' vibration-rotation level, the factor *2(2J'+1)* is the degeneracy of that level (including Λ-doubling) and $Q_{v'}(T(z))$ is the partition function (Herzberg, 1950). "

P.5, L.4: "the OH is in LTE" (see comment on P.2, L.10).

*Please see the answer to that comment. Again, here we state that the model assumes that the OH is in LTE to show that a single-temperature Boltzmann population, at the local temperature of the atmosphere, is used to describe the rotational distribution of emission.*

P.5, L.7-8: The statement related to the cool and warm regions should be weakened since a complete separation of the rotational level populations cannot be expected for typical temperature gradients.

*We have amended the sentence since, as the referee says, the low lines are not only due to the cooler regions. The whole sentence now reads:*

"This results in a net spectrum of low rotational lines whose intensities are enhanced relative to the mean temperature of the emission region when emitted from the cooler regions. Accompanying this are high rotational lines whose intensities are enhanced relative to the mean temperature of the emission region when emitted from the warm regions. Each of these is weighted by the VER of the vibrational transition at each altitude, $N_{v'}(z) \cdot \omega_{v'v''} / \tau_{v'}$."

P.5, L.11: "assuming LTE" should be extended by "for the rotational level populations".

*We agree that this addition makes the text more accurate. The complete sentence now reads:*

"As an example, the model described above was used to create a synthetic spectrum of the (7,4) rotational-vibrational band, assuming LTE for the rotational level populations at every altitude level, for mid-July, mid-latitude conditions".

P.6, L.2: "ergs cmˆ-3 sˆ1" are quite old-fashioned units. Moreover, "sˆ-1" would be correct.

*The missing minus sign has been added. We prefer to keep the units in* ergs cm$^{-3}$ s$^{-1}$ *for direct comparison with the units system used with the SABER instrument.*

P.6, L.7-8: The statement is only correct for the plotted phase of the wave. A phase shift of 180 deg would change the effective sign of the temperature gradient in the OH layer.

*This is true. Later in the paper we model all phases to cover such cases. In this figure caption we only refer to the example given in the figure to highlight the general effect. We have changed the sentence to make this clearer and to avoid confusion. It now reads:*

"With the example wave given here, the temperature variation increases the VER of the lower rotational lines at 95 km altitude, and of the higher rotational lines at 80 km."

P.6, L.10-13: In the same way as Eqs. 5 and 6, Eq. 8 needs to be improved. I propose to write I_v'v"J'J", A_v'v"J'J", E_v'J', and E_v'. This is further modified if the electronic substates are introduced (e.g. by adding i' or F'). Finally, Q_v'(T_rot) is also missing here.

*We have updated the equation as suggested. It now reads:*

"$$\frac{I_{v',v'',J'J''}}{2(2J'+1)\cdot A_{v',v'',J'J''}} = Q_{v'}(T)N_{V'} \cdot exp\left(\frac{-\left(E_{v',J'}-E_{v'}\right)}{kT}\right)$$"

P.6, L.13-16: As Eq. 9 is very similar to Eq. 8, the same proposals are relevant. Depending on the changes, the symbols in Fig. 3 need to be adapted.

*We have also updated this equation. It now reads:*

"$$\ln\left(\frac{I_{v',v'',J'J''}}{2(2J'+1)\cdot A_{v',v'',J'J''}}\right) = \ln(Q_{v'}(T)N_{V'}) - \frac{1}{T}\left[\frac{\left(E_{v',J'}-E_{v'}\right)}{k}\right]$$"

*We have also updated the symbols in the Figure (which is now Figure 4):*

[Figure]

P.7, L.1-2: "the lowest three rotational levels": This obviously refers to the F = 1 substates.

*This is correct. Since we have now said specifically that we only look at the F = 1 sub-state (see comments to P4, L26-29), we keep the text here as it was.*

P.7, L.10: The populations are related to P_1-branch lines.

*Since we have now said specifically that we only look at the F = 1 substates (see comments above), we keep the text here as it was.*

P.7, L.16 - P.8, L.1: This is again an equation which should be improved (see comments on previous equations).

*We have updated this equation in the same manner. It now reads:*

"$\ln\left(\frac{I_{v',v'',J'J''}}{2(2J'+1)\cdot A_{v',v'',J'J''}}\right) = \ln(Q_{v'}(T)N_{V'}) - \frac{1}{T}\left[\frac{(E_{v',J'}-E_{v'})}{k}\right] + \frac{\beta}{T^2}\left[\frac{(E_{v',J'}-E_{v'})}{k}\right]^2$"

P.8, L.1-2: It is stated that the beta parameter was used to improve the fit of the population distribution in Fig. 3. However, the actual value of beta is not provided.

*The β factor for this example is 0.019. We have added this to the text. It now reads:*

"The retrieved temperature $T_{NL}$ = 154.9 ± 0.1 K, is the same as $T_{1,3}$ within the fitting uncertainties, with a β factor of 0.019."

P.8, L.28-30: beta is also not given for the scenarios in Fig. 4.

*The wave used for this figure is the same as used before, so the β-factor is still 0.019. We have added this information to the figure caption. It now reads:*

"This is the same wave as presented in Figure 4. The distribution has a β = 0.019."

P.8, L.32-34: "the distribution of integrated rotational line intensities" for the fitting procedure does not appear to be limited in terms of the rotational levels. If high rotational levels contribute to the fit, the low rotational levels will show under populations. In this case, "excess" will be the wrong term.

*The apparent excess population is defined relative to the predicted Boltzmann population of the lowest three rotational lines. This has also been clarified earlier (see comments above). We have also clarified this point in the text here. The text now reads:*

"The apparent excess population is the ratio between the calculated intensity of a rotational line integrated in altitude, and the intensity of that line predicted by a Boltzmann distribution fitted to the distribution of integrated line intensities of the lowest three rotational lines using a single, effective temperature"

P.9, L.10: Excess populations for P_1-branch lines are shown. The excess is probably related to the first three rotational levels. This information should be given in the caption.

*We agree that this should be stated here. The text now reads:*

"The calculated apparent excess population of the OH (7,4) P($J'$) lines relative to the Boltzmann population characterized by the temperature fitted to the lowest three lines."

P.9, L.14-26: The comparison with the measurements of Pendleton et al. (1993) would be less awkward if the population ratios from their Fig. 16 are also plotted in Fig. 4. It should not be a problem to add histograms of different colour or transparency to the two subfigures.

*We agree that this is a sensible addition. We have changed the figure and associated text as follows:*

[Figure]

Figure 5: The calculated apparent excess population of the OH (7,4) P($J'$) lines relative to the Boltzmann population of the fitted temperature to the lowest three lines. Red on the left is calculated with the climatological temperature gradient shown from NRLMSISE-00 above Boulder, Colorado (40.0° N; 105.6° W) in mid-July. Blue on the right is calculated with a wave of $A_w = 30$ K at 90 km, $\lambda_w = 30$ km and a phase that yields the maximum effect, $\varphi_w = 4.9$ rad. This is the same wave as presented in Figure 4. The distribution has a β = 1.9 %. The grey bars represent the measurements ascribed to NLTE effects from Pendleton et al. (1993).

These results can be compared to the findings from Pendleton et al. (1993) in their Fig. 16, for the (7,4) transition above Boulder, Colorado during mid-summer, the same season and transition as presented here. These results are shown as grey bars in Figure 5 (a) and (b)."

P.9, L.22: This is actually the only place where "P_1" is given. This notation will be understood if the related cases are consistently changed in the paper.

*We have removed the 1 subscript, as we now discuss earlier in the paper that we are only discussing the 1 sublevel. The text now reads:*

"For example, Pendleton et al. (1993) reported an apparent excess population of around a factor of 2 for the P($N$=7) (i.e. $J'$=6.5) line."

P.9, L.26 - P.10, L.2: This conclusion sounds more dramatic than it is since OH kinetic models are usually resolved in altitude, i.e. the models implicitly include the temperature gradient effect. Hence, comparisons with ground-based measurements should be trustworthy without any change in the modelling approach.

*We agree models is the wrong word to use here. We are referring to calculations of kinetic thermalisation based on integrated airglow measurements. We have clarified this by changing the word "models" for "studies". The text now reads:*

"Thus, the effect of the atmospheric temperature background has to be considered in addition to NLTE effects whenever the populations inferred from integrated airglow observations of high $J'$ lines are to be used in kinetic thermalisation studies."

P.10, L.5: It is not clear which lines were used for the baseline fit for the derivation of the excess population. It might be helpful to define the term "excess population" at the first occasion and then to state that this definition is used for the rest of the paper.

*We have defined "excess population" in response to the P2, L12-13 comment.*

P.10, L.6-7: "The figure shows the (7,4) transition" sounds a bit strange. The figure shows results related to the (7,4) band. There are other similar sentences in the paper. As I would primarily expect that "transition" describes a spectral line, "band" or "vibrational transition" appear to be clearer.

*We keep the nomenclature of Pendleton et al. (1993), who discuss the (7,4) "transition", as a comparison to our work. We agree however that this sentence is a bit clumsy and have rephrased it as suggested from the referee. It now reads:*

"The figure shows results related to the (7,4) vibrational transition, which is the same transition that Pendleton et al. (1993) used."

P.10, L.9-10: The discussion in Sect. 3 only focusses on the "highest apparent excess population" for waves of a given amplitude and vertical wavelength. It would also be helpful to briefly quantify the average effect and the related variation. As already discussed in the context of the general comments, it is also important to know what the real distribution of temperature gradients for the OH layer is (or equivalently: the true frequency of the different simulated waves) in order to estimate the real impact of the temperature gradient effect. These results should be compared to the true non-LTE effects (as for the two examples in Fig. 4).

*Such random wave motions can vary with latitude, longitude, season and time of day. We have attempted to give references in the paper indicating that the gradients observed here have been*

*observed. However, a comprehensive survey of all wave motions present in the mesosphere and their statistics is well beyond the scope of this paper.*

*However, we have added a short paragraph on the mean effect of waves with different phases as suggested. The mean effect is about 30 % of the maximum, which was presented in the paper, with a range between 20 % and 40 % dependent on the wavelength and amplitude of the wave. The new text is now:*

"Figures 5 and 6 show the results for the phase which creates the strongest apparent excess population. The mean effect as a result of all different phases simulated is independent of the transition and varies between 20 and 40% of the maximum effect presented in Figures 5 and 6."

P.12, L.3: Just writing "The apparent excess population" is incomplete. In fact, it is "The maximum apparent excess population" for the described wave.

*We have changed the sentence as suggested. It now reads:*

"The maximum apparent excess population of the $J'$ = 6.5 upper level for a wave with wavelength 30 km and an amplitude of 30 K at 90 km altitude as a function of vibrational upper level."

P.12, L.10: "the OH in LTE" (see comment on P.2, L.10).

*See the response to that comment. Our simulations only calculate the OH in LTE, regardless of whether or not this happens in the real atmosphere. We changed the text slightly to make this even clearer:*

"Even though the simulations calculated the rotational population distribution of the OH to be in LTE with the surrounding gas at every altitude, the integrated intensities of the higher rotational lines indicate an apparent excess population that could be misinterpreted as contributing to the NLTE effects previously reported (Cosby and Slanger, 2007; Noll et al., 2015; Pendleton et al., 1993)."

P.12, L.18: It depends on the research goal and the related significance of the effect whether the consideration of the effect is really "necessary". Concerning the estimate of the true impact, the paper should be improved.

*We are confident that this paper shows that a considerable part of the ground-based measured excess population of higher rotational lines can be due to the influence of waves and tides. We therefore think it is important to consider this effect whenever rotational distributions are inferred*

*from ground-based airglow observations. This is exactly what the sentence in questions states. We have no intention to imply that these findings are "necessary" for anything other than this specific research goal. We are confident that the impact presented in this paper is a "true impact", as the waves simulated in this paper have been observed before and could therefore be present in any ground-based measurement. Please see also the next comment.*

P.12, L.18: "the rotational distribution of the OH" might be replaced by "OH rotational level population distributions", which sounds better.

*We have changed the sentence as suggested. It now reads:*

"We conclude that it is necessary to consider the temperature profile in order to infer OH rotational level population distributions from ground-based airglow observations."

Technical corrections:

Whole paper: I have identified four different styles for the vibrational level symbol "v" (see e.g. P.5). In several equations, it is even an upper case letter. This should be harmonised. Other symbols should be checked in the same way.

*We could not find the incident where the vibrational level was upper case. In equation (5), the capital V is referring to the volume emission rate rather than any vibrational level and should therefore be capitalised. We have attempted to harmonise the other instances as suggested by the referee.*

P.9, L.2: "rad" should not be in italics.

*We have changed this in the text.*

References:

[revised manuscript text omitted]

---

## Author Response (AR2)

General comments:

The revised manuscript "Modelled Effects of Temperature Gradients and Waves on the Hydroxyl Rotational Distribution in Ground-Based Airglow Measurements" by Christoph Franzen et al. shows significant improvements compared to the first version. I appreciate these efforts. Nevertheless, several minor issues are still present in the paper. In some cases, some information in the response letter should also be added to the paper as this would also be helpful for the general reader. In some cases, my comments might have not been sufficiently clear and things can be overlooked. Before accepting the paper for publication, the remaning issues should be handled.

*We thank the reviewer for the kind words and all the suggestion and help they offered. We have addressed all the remaining points below.*

Specific comments:

P.2,L.8: The term "radiative effects" is not easy to understand as it is very general. For the reader, it would be helpful to add some words to explain the specific impact on OH. For example, one could write "radiative effects (i.e. the emission of airglow photons) on the roto-vibrational level population distribution" or something similar.

*We have changed the text as suggested. It now reads:*

"The term LTE as used here is not technically correct as it does not account for radiative effects (i.e. the emission of airglow photons) on the rotational-vibrational level population distribution."

P.3,L.2: "rotational level distribution" should be changed into "rotational level population distribution". This is my fault as I forgot to add "population" to the corresponding comment in the first review. In this context, note that "rotational population distribution" (P.3,L.15) seems to be another version with the same meaning.

*We have changed the text as suggested.*

P.3,L.2-6: Indeed, the sentence "Any instrument that integrates ..." will still be true if non-LTE effects are considered. The problems are related to the subsequent sentence "That is, the emission ...". Here, it is not clear whether non-LTE effects are considered, although full thermalisation is needed for the statement (as the OH non-LTE effects tend to increase with altitude irrespective of the kinetic temperature gradient). The phrase "even if thermalised at each altitude" (P.3,L.2) does not imply that the subsequent sentences need this precondition. Moreover, it would still be prudent to change "that occurs" to "that preferentially occurs" or something similar. This sentence does not focus on extreme temperature gradients. In general, the temperature difference between "cold" and "warm" regions should not be sufficient to completely separate the rotational level population distributions for both regions. Note that a similar statement in Sect. 2.3 was corrected in a satisfying way.

*We decided to delete the sentence, which starts with "That is, the emission…" altogether, since it is not necessary in this context and was causing misunderstandings and confusion. We hope, this is a satisfactory solution for the reviewer.*

P.4,L.1: The title "The OH model" is confusing as it suggests that the full model including the rotational level population distribution is described. However, the latter is somehow hidden in the section "Simulation of a ground-based measurement" (Sect. 2.3). Sect. 2.1 is only focussing on the height-dependent vibrational level population distribution, which is necessary for the v'-specific VER profiles. It would be helpful if this was better communicated in Sect. 2.1 by changing the title, giving a brief information on the purpose of this section, and providing a reference to Sect. 2.3 if the structure of Sect. 2 is not changed.

*The OH model section now includes the approximation used throughout the paper, that the OH rotational population is taken to be a Boltzmann distribution characterized by the local temperature. Page 4, lines 25 to 27 now read:*

"As previously mentioned, the rotational population distribution within this vibrational level is taken to be a Boltzmann distribution characterized by the local temperature at this altitude."

*We also added an explanatory sentence at the end of section 2.1., which reads:*

"The application of this model to the simulation of a ground-based measurement under the influence of a realistic temperature gradient is undertaken as described below."

*Lastly, we changed the title of section 2.1 to "The OH steady state model", to clarify the content of this section.*

P.4,L.13: A brief note on the reasons for the neglection of $N_2$ as collision partner would still be useful. This information should not only be provided to the reviewers.

*We added this information to the manuscript. It now reads:*

"$N_2$ is not considered as a quencher in this model. The rate coefficient for OH quenching with $N_2$ is small, and Knutsen et al. (1996) were only able to provide an upper limit. Since the $O_2/N_2$ mixing ratio is nearly constant up to the turbopause, the $O_2$ is three times more effective at quenching than the $N_2$. Thus, neglecting the $N_2$ is well within the uncertainty of the $O_2$ rate coefficient and does not significantly affect the altitude distribution of the OH."

P.5,L.20: Similar to the previous comment, some information on the choice of the Einstein coefficients in the response letter could also be added to the paper. It could be said that the selection of (Langhoff et al., 1986) does not matter for the purpose of this study.

*We assume the referee meant P. 4, L. 20. We added this information to the manuscript. The end of page 4 now reads:*

"Although transition probabilities from a number of different studies differ from those of Langhoff et al. (1986), the integration over the rotational distribution, required to obtain the vibrational

population, is relatively insensitive to the choice of transition probabilities."

P.6,L.8-9, P.7,L.9, P.8,L.3-4, P.8,L.12: In Sect. 2.4, it is not clear which P-branch lines are considered. There is a note on the only use of lines related to F = 1. However, Fig. 3 shows $P_1$- and $P_2$-branch lines, which are called "the P-branch" in the figure caption. On the other hand, "P-branch" is also stated in the caption of Fig. 4, but there it only refers to $P_1$-branch lines. Hence, in Sect.2.4, it should be made clear again which lines are discussed.

*We have added an explanatory comment in the figure caption.*

"Note that even though the figure shows the $P_1$ and $P_2$ lines, only the $P_1$ lines are considered in the model."

P.11,L.17-19: "above about 20 km" is ambiguous as it is not clear where the upper altitude limit is. This is an issue as the statements in the sentence become partly wrong at altitudes of about 30 km depending on the amplitude.

*We have changed this sentence to correct and clarify this point. It now reads:*

"Fig. 6 (a) shows that for vertical wavelengths above about 20 km, the non-linearity of the temperature fit increases with wave-strength, approaching 2% of the linear temperature variation (see Equation 10). The maximum non-linearity is observed at vertical wavelengths of about 20 km for waves with an amplitude of below 10 K, while for waves with an amplitude of 40 K the maximum non-linearity is observed at vertical wavelengths around 40 km."

Technical corrections:

P.6,L.12 and Fig.4: The Boltzmann constant "k" in Eq. (7) is not introduced. Moreover, the abscissa of Fig. 4 shows "k_B", which is inconsistent with Eqs. (7), (8), (9), and (10).

*We have changed all appearances of "k" to "$k_B$" for consistency and added an explanation of the constant below equation (7). It reads:*

"$k_B$ is the Boltzmann constant."

P.7,L.14, P.8,L.1, P.9,L.1: The use of $N_V'$ in Eqs. (8), (9), and (10) still looks wrong. It is not introduced in the paper. However, the entire text (including the sentences directly related to the equations) discusses $N_v'$, which is defined in Sect. 2.1 (P.4,L.19-20). Moreover, $N_v'$ is also used in Eq. (7), which is linked to the equations with $N_V'$.

*We are sorry that these have escaped us in the previous round. We have changed the N_V' in equation (8), (9) and (10) to N_v'. We also note that in the explanation for equation 7, $N_v$ was used instead of $N_{v'}(z)$. This has now been changed.*

General comments:

Overall the authors did a good job replying to the comments in my earlier review. In my opinion the manuscript is close to being acceptable for publication. I have some more minor comments and ask the authors to consider them.

*We thank the reviewer for the kind words and all the suggestions and help they offered.*

Specific comments:

Page 1, line 21: The Meinel papers are probably not the best reference for mesospheric remote sensing using the OH emissions.

*We have changed the citation to Smith et al. (2010), which better fits the context. The text now reads:*

"The hydroxyl (OH) airglow has been employed for many years for remote sensing of the Mesosphere and Lower Thermosphere (MLT) region, an example of which may be found in Smith et al. (2010)."

Page 1, line 31: Add space in ").Low"

*We have changed the text accordingly*

Page 2, line 6: "these two conditions"

It's not clear or obvious what "these two conditions" refers to. I suggest mentioning this explicitly.

*We now mention these two conditions explicitly. The text now reads:*

"In keeping with the terminology employed by Pendleton, Espy, and Hammond (1993), we refer to the condition where the population can be described by a single-temperature Boltzmann distribution as Local Thermodynamic Equilibrium (LTE), whereas Non-Local Thermodynamic Equilibrium (NLTE) is used when the population departs from that distribution."

Page 2, line 12: Add space in ").Here"

*The text has been changed accordingly. (We assume that the referee meant ").The" in line 12).*

Page 2, line 15: ".."

*The text has been changed accordingly.*

Page2, line 20: ".."

*The text has been changed accordingly.*

Page 3, line 15: suggest to replace "i.e. a single Boltzmann distribution" by "i.e. can be described by a single Boltzmann distribution"

*The text has been changed accordingly.*

Page 3, line 16: ".."

*The text has been changed accordingly.*

Page 4, line 20: "its lifetime .. are calculated" -> "its lifetime .. is calculated"

*The sentence has been corrected. It now reads:*

"The total VER of any $v'$ to $v''$ vibrational transition is then given by $V_{v'v''}(z) = A_{v'v''} \cdot N_{v'}(z)$, where $A_{v'v''}$ is the transition probability for the vibrational transition from $v'$ to $v''$ that is calculated from the Einstein coefficients from (Langhoff, Werner, & Rosmus, 1986)."

Page 5, line 9: "the dry adiabatic lapse rate of -10 K/km"

The lapse rate is defined as $-dT/dz$, not $dT/dz$. I suggest sticking to this convention, otherwise your use of "exceed" above (line 1 of this page) is not correct. The figure doesn't have to be changed, but the correct sign should be used when using the term "lapse rate".

*We have changed this (and the following occurrence in the figure caption) to 10 K/km.*

Page 6, equation (6) and line below: The Einstein coeffcent in the line below eqn. (6) does not have the vibrational quantum numbers as indices. I'm also wondering, whether the equation is correct. If the Einstein coefficient also „includes" the vibrational transition, then the 1/tau term should not be required.

*The reviewer is correct, we have revised and corrected the equation. We have also added the vibrational quantum numbers, as advised. The text now reads:*

"The VER of a rotational transition for the $J'$ to $J''$ state from the upper state, $v'$, of the $v'$ to $v''$ vibrational band is given by:

$$V_{v'v''J'J''}(z) = N_{v'J'}(z) \cdot A_{v'v''J'J''}$$

(1)

where $A_{v'v'',J'J''}$ is the transition probability for the $J'$ to $J''$ rotational transition in the $v'$ to $v''$ vibrational transition. Due to vibrational-rotational coupling, these coefficients are specific to each vibrational transition."

Page 6, line 19: Shouldn't the intensity I also include indices for the vibrational states?

*We have changed the text accordingly, it now reads:*

"The VER of each rotational line of a given vibrational transition is integrated through the layer from 74 km through to 110 km to give the intensity of the line, $I_{v'v'',J'J''}$."

Page 7, upper right panel: The spectrum is shown for a temperature of T = 120 K. But T is more like 140 K than 120 K at this altitude, looking at the left panel of the Figure.

*The referee is right, the spectrum was from an older version of the manuscript. We have updated the figure with the correct value of 130 K.*

Page 11, line 25: "can cause up to 30 % apparent excess populations"

This understanding of "apparent excess population" differs from your definition of this term. You defined apparent excess population as the ratio … (bottom of page 9), i.e. the above sentence should read "up to 130% excess population", which – of course – sounds strange. I find your definition of "apparent excess population" not very intuitive, because an excess population of 1 or 100% does not actually correspond to an enhanced population (compared to LTE conditions). In this sense the use of "excess population" is misleading. You did define the term above, but I recommend considering changing the meaning of the term or the terminology. This affects many sentences of the manuscript. I leave it up to you to decide.

*We have taken the referee's suggestion and kept the definition as it is. We now use the apparent excess population according to this definition (several places in the text), and refer to percentages only when describing the percentage of the effect Pendleton et al. (1993) observed that can be accounted for by the temperature gradients. The definition on page 9 now reads:*

"The apparent excess population is the ratio between the intensity of a rotational line integrated in altitude, and the intensity of that line predicted by a Boltzmann distribution fitted to the distribution of integrated line intensities of the lowest three rotational lines using a single, effective temperature. Thus, an apparent excess population of 1 is the same population as predicted from a single-temperature Boltzmann distribution fitted to the lowest three rotational lines. Similarly, an apparent excess population of 2 is a population twice as large as that predicted from this single-temperature Boltzmann distribution."

Page 12, line 12: "Fig. 6 and 7 show the results for the phase which created the strongest apparent

excess population."

This means, it's a different phase for the each of the different combinations of wavelength and amplitude, right? Perhaps this can be mentioned explicitly.

*We did not mention this fact explicitly. The text now reads:*

"Fig. 6 and 7 show the results for the phase that created the largest apparent excess population. Hence, the phase is not constant for each point in these two figures, but rather varies to show the largest apparent excess population for the wave amplitude and wavelength in question."

Page 13, line 1 – 5: Perhaps you would like to mention the reason for the described behavior? The reason is that the internuclear distance is larger for higher vibrational levels, i.e. the molecule's moment of intertia is larger and therefore the value of the rotational constant is smaller, hence the spacing between the rotational levels.

*We added this explanation into the text. The whole paragraph now reads:*

"Tests showed that the difference in the apparent excess population between the (3,1) and (7,4) bands is unlikely to be due to the altitude separation of the different vibrational levels in the OH airglow layer (von Savigny, McDade, Eichmann, & Burrows, 2012). Repeating the analysis and weighting the $v'$=3 and 7 levels with the same VER profile yields essentially the same result, as did performing the analysis in an unperturbed isothermal background temperature profile. Instead, the difference in the apparent excess population between the (3,1) and (7,4) bands is likely due to the compressed rotational energy structure of the higher vibrational levels that lie closer to the dissociation limit. This compression of the rotational energy levels is due to the increased moment of intertia, and hence the reduced rotational constant, associated with the larger average inter-nuclear distance of the higher vibrational levels. Thus a given $J'$ level in a low vibrational state will have more rotational energy than one in a high vibrational state. Thus, for a given temperature, higher rotational levels will be thermally populated in the higher vibrational levels. These thermally populated higher rotational levels then make the perturbing effects of waves relatively less important."

Page 14, line 6: "can explain between 12 % and 30 %"

I suggest mentioning what cases/emissions these values refer to.

*These values refer to the (7,4) transition, which was observed by Pendleton et al. (1993). The case of the 12 % was referring to a wave of 30 K amplitude and 30 km vertical wavelength, as shown in Figure 5. The 30 % case is for the largest waves studied here, as seen in Figure 6. This has been explained more clearly. The text now reads:*

[revised manuscript text omitted]